# A basic framework to explain splice-site choice in eukaryotes

Craig I. Dent[1,11,14], Stefan Prodic[1,12,14], Aiswarya Balakrishnan[1,13,14], Aaryan Chhabra [1], James D. G. Georges[1], Sourav Mukherjee[1], Jordyn Coutts[1], Michael Gitonobel[1], Rucha D. Sarwade[1], Joseph Rosenbluh [2], Mauro D'Amato [3,4,5], Partha P. Das [6,7], Ya-Long Guo [8], Alexandre Fournier-Level [9], Richard Burke [1], Sridevi Sureshkumar [1], David Powell [10] & Sureshkumar Balasubramanian [1] ✉

Changes in splicing can mediate phenotypic variation, ranging from flowering time differences in plants to genetic diseases in humans. Splicing changes occur due to differences in splice-site strength, often influenced by genetic variation and the environment. How genetic variation influences splice-site strength remains poorly understood, largely because splice-site usage across transcriptomes has not been empirically quantified. Here, we quantify the use of individual splice-sites in Arabidopsis, Drosophila and humans and treat these measurements as molecular phenotypes to map variation in splice-site usage through GWAS. We carry out more than 130,000 GWAS with splice-site usage phenotypes, cataloguing genetic variation associated with changes in the usage of individual splice-sites across transcriptomes. We find that most of the common, genetically controlled variation in splicing is *cis* and there are no major *trans* hotspots in the three species analyzed. We group splice-sites based on $GT[N]_4$ or $[N]_4AG$ sequence, quantify their average use, develop a ranking and show that these hexamer rankings provide a simple and comparable feature across species to explain most of the splice-site choice. Transcriptome analyses in several species suggest that hexamer rankings offer a rule that helps explain splice-site choices, forming a basis for a shared splicing logic in eukaryotes.

RNA splicing is a critical gene regulatory process in which specific regions of the pre-mRNA (introns) are removed with the joining of the adjacent regions (exons) to produce the mature mRNA that encodes the protein. RNA splicing affects growth, development, and response to external stimuli[1,2]. Changes in splicing can either alter the protein produced or trigger nonsense-mediated mRNA decay (NMD), regulating the amount of functional protein[3]. RNA splicing can differ within an individual between cells, tissues, organs, and in response to developmental and environmental cues[2]. Tissue-specific splicing has been extensively studied and is known to play crucial roles in diverse biological processes[2,4,5]. In addition, genetic variation can cause splicing changes between genotypes and species as well as genotype-dependent differences in tissue/condition-specific effects, ultimately shaping phenotypic diversity with evolutionary consequences[6–10]. Changes in RNA splicing are also observed in various diseases, including cancer[11,12] demonstrating the importance of splicing in maintaining normal homoeostasis. Thus, there is an interest in understanding the mechanisms through which genetic variation can affect RNA splicing and disease[1].

A vast majority of introns (U2-type) contain a consensus GU at their 5' end (donor) and an AG at their 3' end (acceptor) and are typically processed by the major spliceosome, with a small proportion of introns (U12-type) harbouring alternative motifs (5' GC or AT and 3' AC) processed through the minor spliceosome[13–16]. In addition, a branch point "adenosine (A)" and a polypyrimidine stretch towards the 3' end of introns are known sequence features required for splicing. A rich body of work shows how splicing is an interplay between *trans*-acting factors that are expressed in tissue-specific manner and *cis*-regulatory elements[4,17]. There are intronic and exonic splicing enhancer or silencer motifs at the sequence level in *cis* and RNA binding proteins such as SR proteins and hnRNPs in *trans* modulate the efficiency of splicing[1,4,17–24]. These studies reveal complex regulatory networks governing splicing decisions in a context-dependent manner.

In an RNA molecule, there are multiple GU/AG sequences that could be used in a condition-, tissue/cell type-, or genotype-dependent manner. The rules that govern which of the competing GUs or AGs become splice-sites are critical to understand splicing decisions but remain incompletely described. The ability of a splice-site to participate in a splicing reaction is defined as its "splice-site strength"[25]. Both genetic and environmental variation can influence splice-site strength and lead to differential splicing. While there have been efforts to predict splice-site strength from sequence[26,27], the accuracy of these predictions remains unknown, since splice-site usages are yet to be empirically quantified. Global analysis of splice-sites in various genomes has revealed consensus sequences, and massively parallel splicing assays have been performed to assess the impacts of sequence variation in consensus sequences[1,13,28–30]. For example, at the 5' splice donor site, frequency-based analysis suggests MAG | GTRAGT as the consensus sequence[1,13,28–30]. It is also well known that, beyond 5' GT, sequences within this consensus are extremely diversified and even a few specific dinucleotide combinations that differ between species affect splice-site choice[31,32]. Probabilistic models based on maximum entropy to predict mutational impacts have also been developed[26,33].

Since sequence variation can affect splicing, there is considerable interest in predicting the impact of genetic variation on splicing. Programmes such as SpliceAI, Pangolin, SpliceVault, SPLAM and others use various artificial intelligence (AI) methods to accurately predict the impact of genetic variation on splicing[34–39]. However, it is often difficult to comprehend the biological parameters through which AI-based approaches achieve that high predictive power[40]. Further, the performance of models trained on one species is low when applied to other species[37], which suggests that evolutionary patterns in splice-site choice remain poorly understood.

Accurate measurement of phenotypes is critical for successful and trustworthy genetic mapping. However, efforts to map splicing variation are limited by current measurement methods, which often group multiple splice-sites together, thereby reducing phenotypic accuracy. For example, splicing QTL (sQTL) analyses have been conducted on diverse datasets[41–48], but the quantification of splicing often does not focus on individual splice-sites, which prevents deciphering a direct association between the genetic variant and a specific splice-site[46,47,49]. In addition, sQTL approaches often test only SNPs within a fixed region around the genes as opposed to genome-wide association, typically increasing the risk of spurious associations[41,46,47]. Since most of these studies have focused on regions surrounding splice-sites as opposed to a genome-wide analysis, the question of whether most splicing variation is regulated by *cis* or *trans* genetic variation remains unanswered.

We have previously defined a measure of splicing, Splice-site Strength Estimate (SSE), which focuses on the usage of individual splice-sites[49] rather than general features such as splicing events, isoforms, exon inclusions, intron clusters, or localized splice graphs[48–53]. SSE can be accurately measured with SpliSER (Splice-site Strength Estimate from RNA-seq), and this quantitative measurement can be used as a phenotype in GWAS to detect robust associations[49]. Recent AI-based tools that predict the impact of genetic variation on splicing such as Pangolin[34], SpliceBERT[54] and SpliceTransformer[55] use SpliSER to obtain empirical quantification of splice-site usage and then use this data as training datasets for building their models that outperform AI methods without such training[34,55].

Here, we quantified individual splice-site usage leveraging RNA seq data from: (i) the 1001 Genomes Project from *Arabidopsis thaliana*[56,57], (ii) male and female *Drosophila melanogaster* lines from the DGRP[58,59], and (iii) human heart tissue from the GTEx project[60,61]. We show that there is extensive genetically controlled variation in individual splice-site strengths. By carrying out more than 130,000 GWAS using these data, we catalogue both *cis* and *trans* genetic variation influencing individual splice-site usage across the genomes in Arabidopsis, Drosophila and humans. We demonstrate that most of the variation in splice-site strength/usage is driven by *cis* regulatory variation. With high-resolution GWAS that map around splice-sites, we infer specific nucleotides for each position that are best suited to promote splicing and identify features that govern splice-site choice. We subcategorise splice-sites based on their intronic hexamer sequences encompassing splice donor and acceptor sites, compute their average strengths and rank the hexamers based on their average strengths. We show that the ranking of the average strengths of hexamers correlates with their presence at splice-sites. We present hexamer rankings as a minimal sequence-level feature that explains most splice-site choices in the transcriptomes of diverse species. Our findings suggest that hexamer ranking presents a basic logical framework that explains most splice-site choice across eukaryotic organisms. This framework, while not capturing all regulatory layers, nonetheless highlights a conserved sequence-level logic that may be further modulated by context-specific regulatory inputs.

## Results

### Extensive variation in splice-site usage in Arabidopsis, Drosophila, and humans

To assess splicing variation, we first quantified the usage of all splice-sites from population-level panels of Arabidopsis[57], Drosophila[62] and Humans[60] with SpliSER[49] and obtained splice-site usage data for a total of 767,199 splice-sites representing 39,734 genes from three different species (Supplementary Table 1). Across all genotypes in the three species, this represented the quantification of splice-site strength for ~200 million splice-sites. Since researchers have been using tools such as the MaxEnt score that predict splice-site strength, particularly in humans[26], we assessed how these predictions reflect empirically quantified splice-site strengths. We observed that empirical quantifications differed noticeably from MaxEnt predictions (Supplementary Fig. 1A, B). The distribution of MaxEnt scores, for both donors and acceptors, displayed a skew towards the higher range, which suggests that MaxEnt, while capable of differentiating splice-sites from non-splice-sites, is less effective at reflecting observed usage, which could be context-dependent. Comparison of the empirically quantified splice-site strengths between different tissues (heart vs testis) (Supplementary Fig. 1C) and sex (Drosophila) (Supplementary Fig. 1E) showed the potential to identify sites with differential usage in distinct contexts, distinguishing the empirical quantifications by SpliSER from sequence-based predictions such as MaxEnt.

Next, we computed: (a) the range of splice-site usage (difference between the highest and lowest SSE) in individuals, and (b) the variance in the usage of every splice-site. We observed extensive variation in the usage of individual splice-sites. Almost three-quarters of the human sites displayed more than 20% difference in their usage between individuals, while a half (50%) and a quarter (25%) of the sites showed such differences in Drosophila and Arabidopsis, respectively (Fig. 1, Supplementary Table 1). In summary, we observed 430,345 splice-sites (56.1%) out of a total of 767,199 that displayed more than a

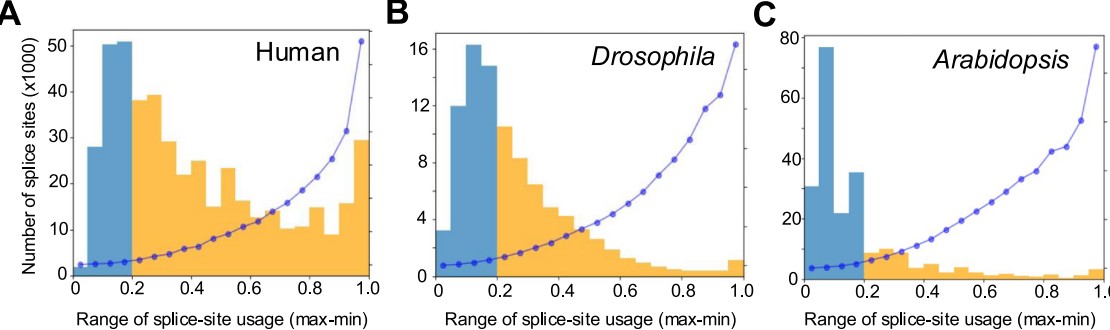

**Fig. 1 | Splice-site usage varies extensively between individuals.** Distribution of splice-sites that show differences in their usage between individuals in humans (**A**), Drosophila (**B**) and Arabidopsis (**C**). The sites are grouped based on the maximum difference in their usage between individuals. The yellow region highlights the sites that display more than 20% difference (0.2 difference in) between extreme samples. Blue dots represent the average variance in splice-site usage between all individuals for each of the bins.

20% difference between the highest and the lowest usage among individuals (Supplementary Table 1).

The transcriptomes from Arabidopsis[57] and Drosophila[58] had replicates but human samples lacked replication. To assess whether the lack of replication could account for the increased variability seen in human samples, we compared the splice-site usage of two different skin tissues (leg vs suprapubic region) (Supplementary Fig. 1D). There was a high degree of concordance in data ($R^2 = 0.99$) bolstering confidence in our quantifications. We exploited Arabidopsis and Drosophila replicates to calculate broad-sense heritability of splice-site usage and found that a substantial number of sites (>25%) displayed high heritability (>0.3 in Arabidopsis and even higher in Drosophila). This suggested that there is a substantial genetic contribution to splicing variation that could be mapped. To maximise the probability of mapping by GWAS, and to avoid spurious associations, we selected all splice-sites for which there are at least 100 observations and that are in the upper quartiles of both variance and heritability, which resulted in a total of 17,140 sites in Arabidopsis, 7711 sites in the Drosophila female and 8003 in the Drosophila male datasets. For humans, we took all 97,796 sites that had data for at least 100 individuals and fell in the upper quartile of variance. In total, we took splice-site strength data from 130,650 sites for subsequent GWAS (SpliSER-GWAS).

## SpliSER-GWAS is specific and captures causal variants for differential splice-site usage

Using SpliSER-GWAS, we detected associations for a total of 21,988 sites (4951 in Arabidopsis, 7328 in Drosophila and 9709 in Human; Supplementary Data 1-3). To assess the ability of our approach to identify causal variants, we took advantage of mutations at splice-sites, which will have a causal impact on splice-site usage and should be reflected in GWAS. Such mutations often lead to usage of an alternative site. We reasoned that we should identify the splice-site mutation as the highest associated SNP for both the mutated site and the alternative site. For example, at the *TOR1AIP1* locus in humans, there are individuals with a G to A mutation at the splice acceptor site at chr1:179,889,309. In individuals with this mutation, another acceptor site at chr1:179,889,312 is used (Fig. 2A). We identified the G to A mutation at chr1:179,889,309 as the highest associated SNP for variation in the usage of both sites (chr1:179,889,309 & chr1:179,889,312) (Fig. 2B), while the other sites on the same gene were unaffected. We observed similar examples in Drosophila (Supplementary Fig. 2A) and Arabidopsis (Supplementary Fig. 2B).

To assess the extent to which we can capture causal variation, we analyzed all splice-site mutations that occur in at least 5% of the individuals by GWAS (minor allele frequency ≥0.05). This resulted in a total of 58 sites in Arabidopsis, 63 and 34 sites in Drosophila males and

females, respectively, and 48 sites in Humans (Supplementary Table 2). 152 of these 203 sites gave clean GWAS peaks with negligible noise (~75%; 43/58 in Arabidopsis, 53/63 in Drosophila males, 27/34 in Drosophila females and 29/48 in Humans). More importantly, we identified the splice-site mutation as the highest and/or closest associated SNP in 34/58 sites (59%) in Arabidopsis and 29/48 sites (60%) in human heart tissue, 36/63 (57%) in Drosophila males and 16/34 (47%) in Drosophila females. These findings suggest that SpliSER-GWAS can capture up to ~60% of the mappable causal variants with large-effect.

Next, we compared SpliSER-GWAS results with previously published sQTL analysis from the same dataset[46]. The key aspects that differentiated our study include splice-site level-specificity of the phenotype and a more conservative statistical threshold (Supplementary Table 3). In addition, sQTL analysis reported all significant SNPs, while we report only the highest and closest associated SNPs for each phenotype. We noted that 10 of the 39 mappable splice-site mutations (25%) were found to be the highest associated SNPs for their corresponding sQTL phenotypes. However, 82% (32/39) were found to be among significant (though not the highest associated) SNPs, indicating a reduced signal/noise ratio. We then compared the associations at the level of genes. We observed a significant (hypergeometric probability $p = 4.45e^{-443}$) overlap of 40% in genes for which splicing variation is mapped by both sQTL analysis and SpliSER-GWAS (Supplementary Fig. 3A). The distribution of *p*-values suggested that SpliSER-GWAS *p*-values were generally lower than sQTL p-values (Supplementary Fig. 3B). Several genes for which only sQTL analysis found an associated variant were either not considered to be variable based on our phenotypes, or they failed to pass the stringent thresholds applied in SpliSER-GWAS (Supplementary Fig. 3C). On the other hand, genes that were only detected by SpliSER-GWAS produced clear GWAS signals, suggesting the potential to capture and map variation previously missed by sQTL analysis, perhaps due to the differences in the phenotypes used (Supplementary Fig. 3D).

## Most of the genetically associated splicing variation is *cis*

To assess the genomic architecture of splicing variation, we plotted the positions of splice-sites against their highest associated variants in all three species (Fig. 2C–E). We observed that most of the determinants of splicing variation mapped within 1 Mb from the splice-site in all three tested species (Supplementary Data 1–3). We denoted these as *cis* and the rest as *trans*. Cis-associations represented 78-91% of all associations (Table 1), confirming that most genotype-dependent splicing variation is mediated via *cis*- rather than *trans*-genetic variation in all three species. We observed that 40–60% of the unique associated SNPs were in the same gene (1564/4013 in humans, 2005/4155 in Arabidopsis and 2592/4234 in Drosophila) that harboured the

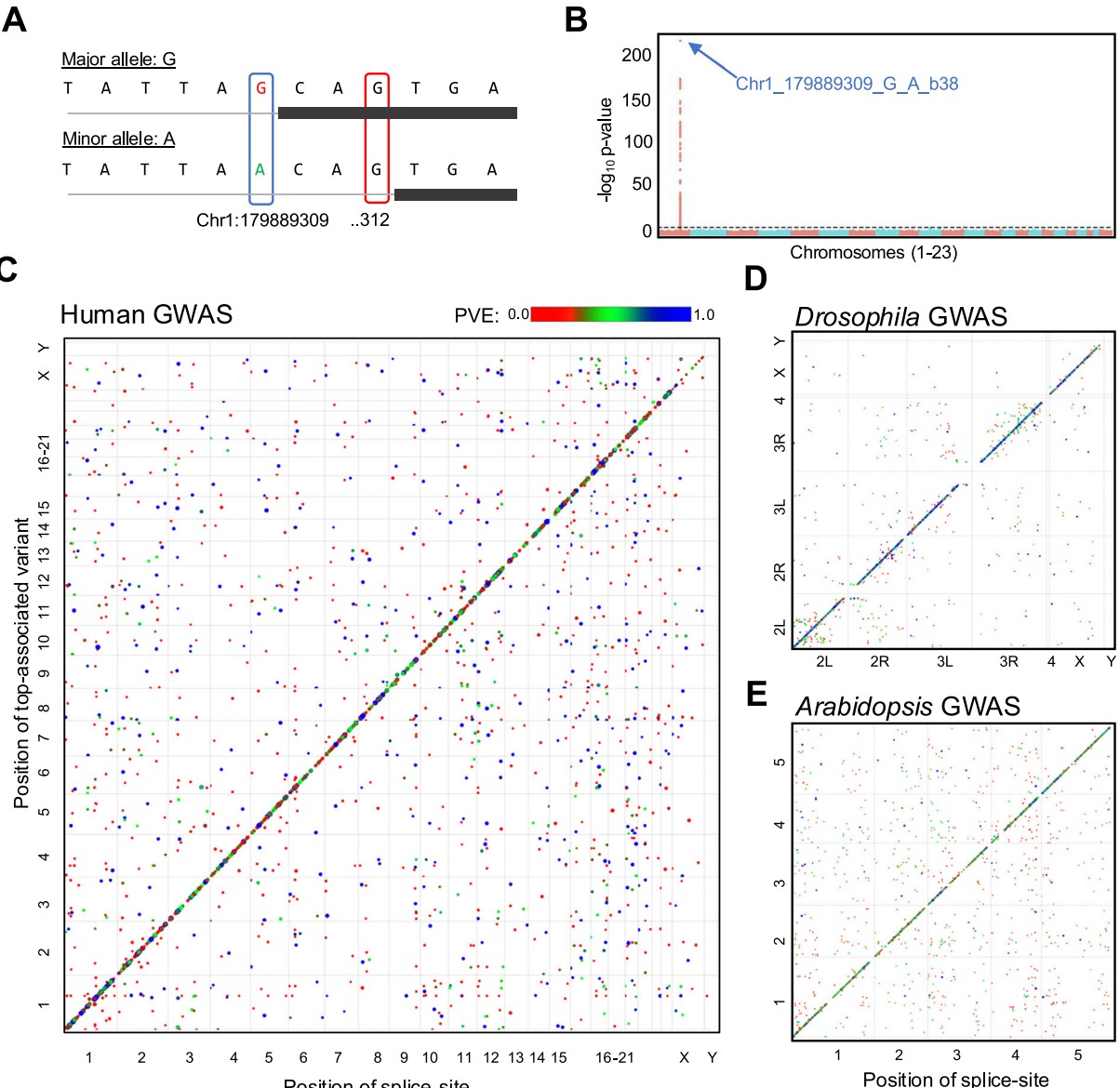

**Fig. 2 | Splicing variability can be mapped accurately, and it is mostly *cis*-regulated. A** A schematic of the sequences surrounding two competing splice-sites in *TOR1AIP1* is shown. **B** Manhattan plot of the splice-site mutation at chr1:179,889,309. SpliSER-GWAS analysis identifies causal SNP for variation in the usage of splice-sites at the *TOR1AIP1* locus in humans. The splice-site mutation at chr1:179,889,309 (309) allows the usage of chr1:179,889,312 (312) as a splice-site,

and variation in the usage of both sites map to the 309 polymorphism. Scatter plot of splice-site positions and their highest associated SNPs in the human heart (**C**), Drosophila (**D**) and Arabidopsis (**E**) across corresponding genomes. Colour scale represent the proportion of variance explained (PVE) by the associated top SNP. The sizes of the dots are also correlated with PVE.

splice-site. Of these ~50–60% SNPs (728/1564 in humans, 952/2005 in Arabidopsis and 1567/2592) were in the same exon/intron as the splice-site.

## There are no major *trans* hotspots of common allelic variation affecting splicing

Most previous sQTL analyses lacked the ability to map *trans* associations since they focused on variants in regions surrounding the genes[45,46,48,63] (Supplementary Table 3). In contrast, our study is an unbiased genome-wide association analysis, which allowed us to assess the effect of *trans*-genetic variation on splicing. Though we observed that the majority of variation is *cis*, we also mapped *trans* associations (Table 1). We observed that human GWAS identified a maximum of ~22% (2161 out of a total of 9872) of the associations being *trans* associations. This was similar in Arabidopsis (916 out of a total of 5277,

17%) and somewhat reduced (678 out of 7791, 9%) in Drosophila. Since mutations in splicing factors are often found in diseases including cancer[64], we asked whether any of our highest associated SNPs in *trans* peaks fall on genes encoding spliceosomal proteins or other RNA-binding proteins. We reasoned that if there were such an association, we would observe variation in multiple splice-sites mapping to such variants and thus they may present hot spots. However, we did not observe any major *trans* hot spots (Fig. 2C–E).

Unlike the *cis* SNPs where the proximity to splice-site could offer some confidence on potential causality, without experimental analysis, it is difficult to evaluate the causality of *trans* SNPs. In addition, duplicated genes can cause mapping issues, which could appear as a *trans*-association, as we observed for the largest effect *trans* peak in Arabidopsis (Supplementary Fig. 4). Therefore, further experimental analysis is required to explore *trans* associations. In the absence of

**Table 1 | A summary of associations detected in different RNA-seq datasets**

| Species/Phenotype | Total associations | *cis* associations | *trans* associations | % of *cis* associations | % of *trans* associations |
|---|---|---|---|---|---|
| Arabidopsis | 5277 | 4361 | 916 | 82.6% | 17.4% |
| Drosophila(male) | 3897 | 3559 | 338 | 91.3% | 8.7% |
| Drosophila(female) | 3453 | 3143 | 310 | 91% | 9% |
| Human heart atrial tissue | 9872 | 7711 | 2161 | 78.1% | 21.9% |

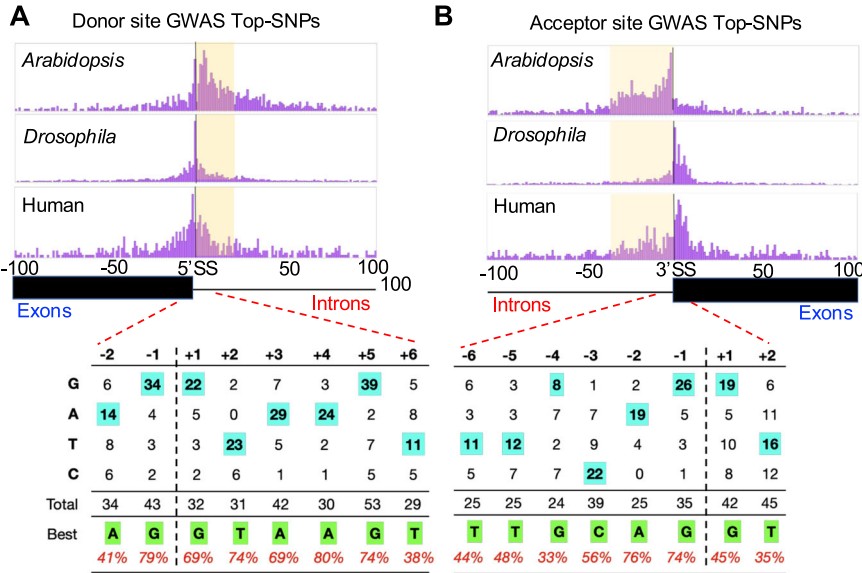

**Fig. 3 | Genetic variation affecting splice-site choice is often near the splice-site and high resolution SpliSER-GWAS allows inferring the best nucleotides that promote splicing.** Distribution of the distances of the highest associated SNPs (lowest *p*-value) detected in SpliSER-GWAS for donors (**A**) and acceptors (**B**) in Arabidopsis, Drosophila and Humans. Intronic regions are shaded for clarity. The distribution of splice-promoting allelic variation for positions −2 to +6 around the splice-donor site and −6 to +2 around the splice-acceptor site based on associations from all three species, considering GWAS where the Top SNP and the closest SNP are the same. Most frequent splice-promoting nucleotides are highlighted.

experimental analysis, we conclude that there are no major *trans* hot spots in any of the three analyzed species.

### Intronic regions drive most of the variation in splice-site usage

To assess the features of splicing variation, we plotted the distribution of distances of the highest and closest-associated SNPs to the splice-site for all sites in all three species, which revealed that most of the associations map in and around the splice-site (Fig. 3, Supplementary Fig. 5A, B). We observed 32% (1410/4361, 611 donors and 799 acceptors) associations in Arabidopsis, where the highest-associated SNP fell within 100 bp of the splice-site; we saw 24% in Drosophila (1771/7350, 772 donors and 999 acceptors) and 9% (651/7711, 281 donors and 370 acceptors) in humans. These numbers almost doubled if we considered the closest SNPs within the peak to the site (60%, 52% and 23% in Arabidopsis, Drosophila, and humans, respectively). We observed an intronic bias in Arabidopsis, but the pattern was more diffused between exons and introns in Drosophila and Humans, which suggested that the genetic architecture of splicing variation may differ between species (Fig. 3). Additionally, the aligner used for Arabidopsis was different. Though all samples aligned with the same aligner clustered together in a PCA plot (Supplementary Fig. 6), suggesting that they may not have a large effect on mapping variation, we cannot conclusively rule out the effect of aligners. Nevertheless, these findings suggest a strong influence of SNPs near splice-sites affecting variation; we also observed that 40-60% of the SNPs fell in exons/introns that were not adjacent to the splice-sites. We asked whether the effect size of the mapped SNPs depended on their proximity to the splice-sites. We observed maximum effect for SNPs that are in the same intron/

exon as that of the splice-site, that on other regions of the same gene (Supplementary Fig. 7).

To assess whether intronic or exonic nucleotides make the biggest difference to splice-site strength, we analyzed the average strength of all splice-sites with specific nucleotides at each position around the splice-site in all three species. We noticed that the strongest and most frequent nucleotides differed more on the exonic side than on the intronic side (Supplementary Fig. 8, Supplementary Data 4 & 5). When we considered pairwise interactions of nucleotides and their impact on splice-site strength, we observed interactions between the -1 position and the +3/+4/+5 positions of the intronic region (Supplementary Figs. 9–12; Supplementary Data 6 & 7), consistent with previous literature[13,65]. Finally, we asked how much of the variation in splice-site strength/usage across the genome can be explained by individual nucleotides surrounding the splice-site by taking all splice-sites into account. This analysis suggests that splice-site strength is mostly affected by intronic rather than exonic sequences (Supplementary Fig. 13).

### High resolution GWAS allows inferring splice-promoting nucleotides

A substantial number of variants were mapped at each position around the splice-site in GWAS. In each of the associations, there are SNPs (alleles) associated with increased (splice-promoting) or decreased (splice-reducing) usage of the splice-site. We inferred the most frequent bases among the splice-promoting (best) and splice-reducing (worst) nucleotides for each position (Fig. 3A, B, Supplementary Fig. 14A–C, Supplementary Data 8). These sequences would be

identical if driven by background nucleotide frequencies. We observed these sequences to differ in general, indicating that these patterns are unlikely to have evolved neutrally.

To assess the functional impacts experimentally, we designed the best and the worst synthetic introns of 100 bp containing the splice-promoting (best intron) and splice-reducing (worst intron) nucleotides at each position (Supplementary Data 9). We noted that, our synthetic best intron was "AT"-rich (75%) and the worst intron was "GC" rich (68%) consistent with literature[66]. We introduced these synthetic introns into the coding region of mCHERRY between AG and GT (AG)-[GTintronAG]-[GT]. Consistent with our expectations, we observed strong fluorescence in HEK293T cells that were transformed with the construct having the best intron (Supplementary Fig. 15A). Cells that were transformed with the worst intron (despite harbouring GT-AG and a branch point) displayed no fluorescence. RT-PCR analysis confirmed the appropriate splicing of the best intron and no splicing of the worst intron, consistent with our prediction (Supplementary Fig. 15B). We conclude that allelic variations beyond the GT/AG and the branchpoint adenosine have a significant impact on splicing and contribute to splicing variation.

### GT[N]₄ and [N]₄ AG hexamer sequences help explain splice-site choices across species

Given that the intronic region around the splice-site primarily accounts for variation in splice-site usage, we attempted to identify the smallest intronic region, comparable across species, that could explain a substantial proportion of splice-site choices. We systematically scanned a 100 bp window (50 bp upstream and downstream) surrounding every individual splice-site for all GT or AG dinucleotides, extracted corresponding $k$-mers (GT, GTN, GTNN, etc.) and queried in what proportion of splice-sites $k$-mer ranking can explain splice-site choice. We determined the percentage of splice-site choices explained by different $k$-mer lengths (i.e., the percentage of sites in which the highest-ranked $k$-mer was being utilized as the splice-site) and found that GT[N]₄ hexamers explain most (70.8%) of the donor-site choices (Table 2). For acceptor sites, we found that both hexamers [N]₄AG and [N]₅AG had similar scores (~60%), though the hexamers on average, slightly outperformed the heptamers across species (Table 2). All three species gave similar patterns (Supplementary Table 4). We conclude that hexamers present an optimal intronic $k$-mer for cross-species comparisons.

**Table 2 | Percentage of splice-site choices explained in three different species with different lengths intronic $k$-mers**

| Sequence | *Arabidopsis* | *Drosophila* | Humans | Average |
|---|---|---|---|---|
| **Donors** | | | | |
| GT | 0.66 | 0.76 | 0.87 | 0.76 |
| GTN | 28.56 | 15.02 | 26.73 | 23.43 |
| GTNN | 38.45 | 48.73 | 52.42 | 46.53 |
| GTNNN | 56.06 | 77.47 | 69.87 | 67.80 |
| **GTNNNN** | **58.26** | **81.41** | **72.86** | **70.84** |
| GTNNNNN | 48.95 | 65.84 | 69.05 | 61.28 |
| GTNNNNNN | 24.60 | 40.90 | 53.24 | 39.58 |
| **Acceptors** | | | | |
| AG | 0 | 1.93 | 0 | 0.64 |
| NAG | 30.24 | 28.46 | 19.49 | 65.19 |
| NNAG | 48.95 | 46.04 | 32.14 | 42.37 |
| NNNAG | 60.98 | 64.05 | 46.46 | 57.16 |
| **NNNNAG** | **65.73** | **71.02** | **55.25** | **64.00** |
| NNNNNAG | 65.71 | 65.03 | **57.72** | 62.00 |
| NNNNNNAG | 53.92 | 43.29 | 51.10 | 49.43 |

$k$-mers that explain maximum variation are shown in bold.

We carried out a similar analysis using MaxEnt in human data. MaxEnt utilises additional sequence information (9-mer for donors and 23-mer for acceptors), which makes it difficult to interpret differences between species. Unsurprisingly, MaxEnt performed similarly with donors (~76% of splice-site choices explained) but outperformed hexamers with acceptors (83% compared to 60%). We then computed the correlation between hexamers and MaxEnt by compressing MaxEnt scores into hexamer groups (Supplementary Fig. 16). We obtained a very strong correlation, particularly with acceptors ($R^2 = 0.9$), which suggests that hexamers disproportionately contribute to majority of the MaxEnt scores (Supplementary Fig. 16A–D).

We observed significant correlations (0.74 for donors and 0.6 for acceptors) between the frequency of hexamers and their average splice-site strength, but not with their frequency in gene bodies or genome (Supplementary Fig. 17). In addition, the proportions of used hexamers in gene bodies were correlated with hexamer strengths ($R^2 = 0.74$ for both donors and acceptors), unlike the proportion of unused hexamers ($R^2 = 0.001$ for both donors and acceptors, Supplementary Data 10 & 11). Taken together, these data indicate that natural selection probably plays a role in choosing different hexamer sequences at the splice-site.

To assess the generality of this observation, we analyzed publicly available RNA-seq data to compute hexamer ranks based on their strength and frequency across 25 eukaryotic species and asked what proportion of the splice-site choice could be accounted for by hexamer rankings. Indeed, hexamer ranking based on splice-site strength explained most splice-site choices (~60-85%) in diverse species (Supplementary Data 12, 13, Supplementary Table 4). Next, we computed pairwise rank-correlations of hexamer rankings and constructed dendrograms based on their $R^2$ values. $R^2$ values in all species comparisons were positive, and reflected phylogenies, to some extent suggestive of evolutionary conservation (Supplementary Data 14, Supplementary Fig. 18A).

### Splicing decisions are driven primarily at the level of species

Our finding that there is evolutionary conservation of the hexamer rankings seemingly contradicts previous findings that the splicing code is tissue-specific[9,10]. To clarify this seeming inconsistency, we ran SpliSER on the same data from four different tissues of four different species from Barbosa-Morais et al.[9] and asked whether the clustering of tissue-specific splice-sites differs from that of other splice-sites. Irrespective of how we split the data, we observed clustering based on species (Supplementary Fig. 19). We also observed stronger correlations within a species for hexamer ranks for both donors and acceptors across all tissues (Supplementary Fig. 20). Thus, species-level clustering seen in previous findings[9,10] and in our work together suggests evolution of the splicing machinery at the species level. This is consistent with a fundamental conserved logic of splice-site choice, with tissue-specificity acting on top of that framework, potentially based on the expression of tissue-specific *trans*-acting factors.

We wondered, for donor sites, if this could be explained purely in terms of U1snRNA base pairing with the 5' splice-sites. The most frequent and the strongest hexamer GTAAGT (Supplementary Data 12) is complementary to the most common U1 snRNA base pairing site of CAUUCA. However, only 23% of splice-sites harbour this consensus sequence. Therefore, we grouped the hexamers using sequence distances from the U1 snRNA binding site (GTAAGT) and asked whether the average strengths of these hexamer groups reflect sequence distance from the U1 snRNA binding site. Sequence distance matrices do not provide the same level of resolution as the average strengths. Nevertheless, we observed a near-perfect correlation between the sequence distances and the hexamer rankings in Arabidopsis, Drosophila and Humans (Supplementary Fig. 18B). This data suggests that, at least for the donor sites, hexamer-based differences in splice-site choices is primarily driven by U1 snRNA base-pairing properties. For

the 3' acceptor sites, we computed the distances from the hexamer with the highest strength (TTGCAG) and found a very similar correlation between sequence distances and hexamer rankings (Supplementary Fig. 18C). Taken together, these results indicate that intronic hexamer sequences potentially form a basic feature of the splicing code in eukaryotes.

## Hexamer ranking explains splicing in both natural and experimental perturbations

**Natural variation.** Since intronic hexamers explained splice-site choices, we tested whether they could explain the effects of previously described mutations. Two different transcripts are produced from the *FLOWERING LOCUS M (FLM)* gene of Arabidopsis based on the splicing of the donor GT of the first intron with AG1 or AG2 acceptor sites. Generally, AG1 is preferred over AG2 and differential splicing of *FLM* leads to changes in temperature-dependent changes in flowering[67–69]. Hexamer analysis revealed that AG1 harbours a stronger hexamer compared with AG2, which can explain its preference (Supplementary Fig. 21A). In addition, a mutation (GG to AG at position chr1:28,958,437) in the second intron of *FLM* abolishes the use of either of these partner sites[70]. It turns out that the mutation creates a new AG acceptor site, which then has the strongest hexamer of the three, which can explain the abolition of the use of AG1/AG2 (Supplementary Fig. 21A).

Similarly, in humans, several mutations in the *CFTR* gene are associated with splicing differences[71]. One of the common mutations is *rs75039782* (C to T) in the 22$^{nd}$ intron of the *CFTR* gene, which generates a new donor GT site[72]. This mutation leads to a pseudo-exon containing a premature stop codon, resulting in CFTR deficiency and cystic fibrosis. Hexamer analysis explains this mutational impact: The *rs75039782* mutation generates a donor site with a stronger hexamer than the canonical donor site, resulting in an otherwise unused upstream acceptor site in the intron now being used in partnership with the canonical donor site that leads to the inclusion of the pseudo-exon (Supplementary Fig. 21B).

**Meta-analysis of large-scale manipulation data.** We leveraged data published by Rosenberg et al.[30] and carried out a meta-analysis with a focus on hexamer ranking[30]. This dataset from cell culture experiments allowed the direct testing of thousands of competing hexamers. We ran SpliSER on this RNA-seq data and quantified the splice-site usage of all splice-sites. A total of 466,000 competing donors allowed us to evaluate ~4000 unique pairs of competing hexamers. We queried how many of the winning splice-sites could be explained by the hexamer rankings and found that hexamer ranking could explain 66% of splice-site choices (Fig. 4A). This percentage increased to 91% when observing constructs where the differences in hexamer strengths were more than 0.25, providing further evidence that hexamers indeed explain splice-site choices (Fig. 4A).

**Mini-gene assays.** We designed three types of mini-gene constructs with the mCHERRY reporter gene. First, we scanned for human introns that are removed with more than 95% efficiency (i.e., donor and acceptor SSE ≥ 0.95, good intron) or less than 10% efficiency (i.e., donor and acceptor SSE ≤ 0.1, bad intron) across all individuals in the GTEx dataset. We reasoned that by changing the hexamers, we could convert bad introns into good ones and vice versa. An intron in the *RPS10* gene with a splice donor site at chr6:34,425,071 and an acceptor site at chr6:34,424,840 is removed with almost 100% efficiency (>99% across all GTEx individuals). We generated constructs where the hexamer sequences at the donor and acceptor sites (GTAGGA [Rank 18, SSE-0.626] / TTACAG [Rank 26, SSE 0.631]) were changed to weaker hexamers (GTCCAA [Rank 241, SSE 0.038] / TGCGAG [Rank 217, SSE 0.037]; Supplementary Data 9, 12,13) and transfected the constructs into HEK293T cells. We analyzed mCHERRY fluorescence and performed RT-PCR (Fig. 4B). Changing the hexamers abolished splicing at

these splice-sites but revealed cryptic 5' and 3' splice-sites that resulted in an inclusion of 26 bp (additional 8 bp from the alternative donor site and 18 bp from the alternative acceptor site), which caused frameshift disrupting mCHERRY fluorescence.

Second, as a weak intron we selected intron 2 of *CALM3*, (chr19:46,608,327-46,608,481) and changed the donor hexamer from GTGGAT [Rank 200, SSE 0.117] to GTGAGT [Rank 3, SSE 0.724]. With the original CALM3 intron, an alternative donor 13 bp downstream (GTGAGC [Rank 9, SSE 0.653]) is used in splicing, which disrupts mCHERRY expression due to a frameshift (Fig. 4C). However, change of the hexamer resulted in proper splicing with mCHERRY fluorescence (Fig. 4C).

Third, we selected *MYO15B* intron (chr17:75592302-75592463) where two acceptor sites at chr17:75592427 and chr17:75592463 compete while the donor is less variable. GWAS analysis showed that the splice-site choice mapped to an SNP at chr17:75592461 (A/G) in the hexamer sequence (TACAAG [Rank-148, SSE-0.279]/TACGAG [Rank 201, SSE-0.049], Supplementary Data 3). We changed the sequences at these competing sites to have various combinations of hexamers with differing strengths (Supplementary Data 12,13). Confirming the causality in our GWAS results, a change of A to G reduced the use of chr17:75592463 and promoted the use of chr17:75592427. Analysis of diverse combinations revealed that the highest mCHERRY expression and proper splicing was observed with the construct where we supplied the donor and acceptor sites with higher-ranked hexamer sequences (Supplementary Fig. 22A, B). In all other cases, we observed sub-optimal splicing, including the use of alternative splice-sites (Supplementary Fig. 22A, B).

Finally, we took our synthetic worst intron and modified it to assess minimal features that could improve splicing. Adding a poly-pyrimidine tract and a branch point with consensus sequences had negligible effects suggestive of additional requirements (Supplementary Fig. 23). When we also changed both donor and acceptor hexamers into stronger ones, we could observe some level of proper splicing (Supplementary Fig. 23). Taken together, these findings experimentally demonstrated that the hexamer sequences are among the primary factors that not only explain splice-site choice but also could be used to rationally engineer desired changes in splice-site strength.

## Discussion

We have demonstrated that using the individual splice-site strength as a phenotype for GWAS specifically links genetic variation to the usage of specific donor or acceptor splice-sites. Often, a mutation that directly impacts the strength of one splice-site will have indirect effects on the usage of other sites. For example, in the case of *MYO15B* (Supplementary Fig. 22A, B), an SNP at chr17:75592461 reduces the efficiency of the splice acceptor site 2 bp downstream of the SNP (chr17:75592463), but also indirectly increases the efficiency of the competing acceptor site at chr17:75592427. We were able to not only demonstrate that we could map splice-site usage of both competing sites to allelic variation at chr17:75596461 (Supplementary Data 3) but also confirm this through minigene assays (Supplementary Fig. 22A, B). Thus, we can differentiate direct and indirect effects in many instances simply from the distance of the associated SNP from the splice-site. Differentiating such primary and secondary effects is harder to achieve with other approaches. Thus, we would argue that our approach, focused on individual splice-sites, would provide a better understanding of the regulation of splicing as opposed to other approaches based on isoforms or splicing events.

We have shown that common splicing variation is primarily driven by *cis* rather than *trans* regulatory genetic variation. While it is clear that there are several *trans*-acting factors that play a critical role in splicing[29], natural variation in *trans* effects (i.e., a change in splicing driven by a *trans* acting factor) could be brought about in two different

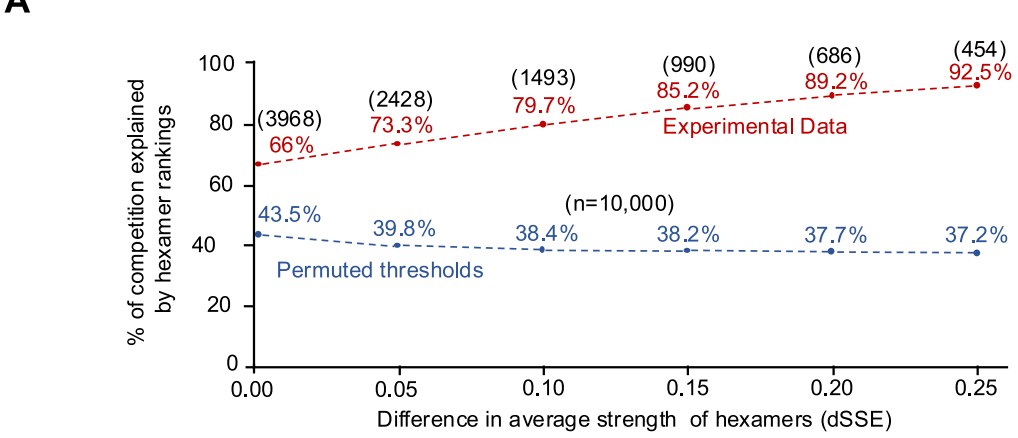

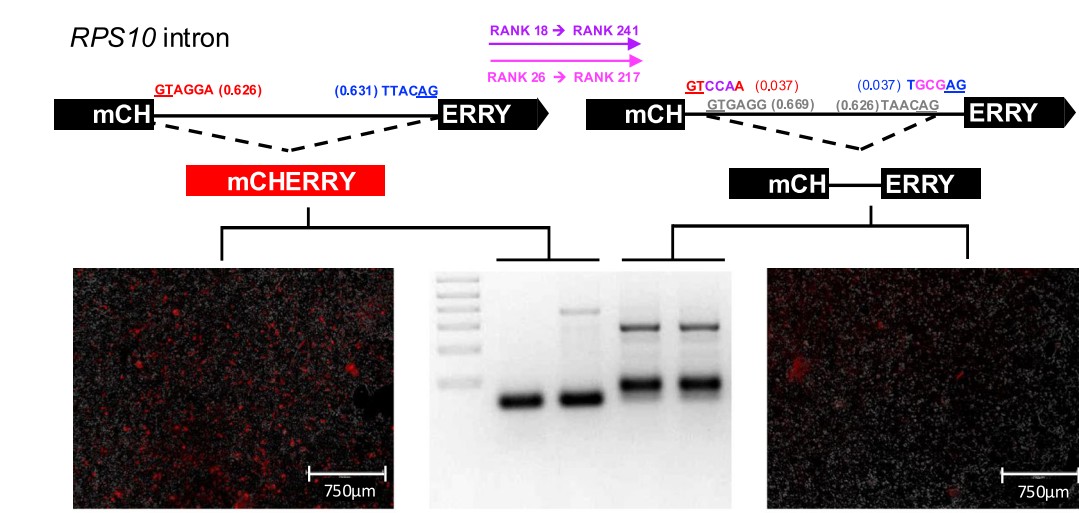

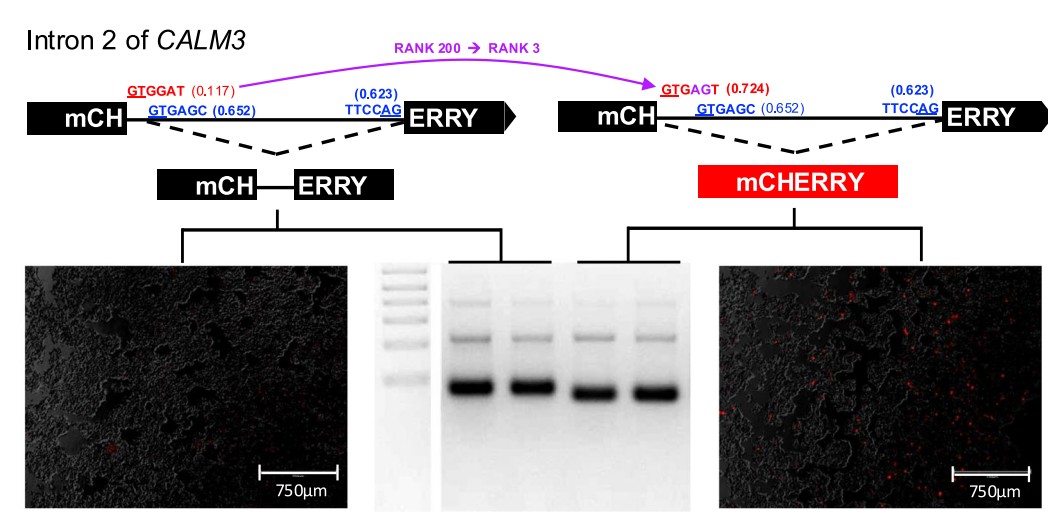

**Fig. 4 | Experimental perturbations of hexamer sequences alter splice-site choice. A** Direct comparison of competing hexamers in minigene constructs of Rosenberg et al.[30]. Blue line represents permuted thresholds calculated from 10,000 permutations and the actual experimental data (red) with varying combinations of hexamers. The number of unique competing pairs tested is shown above the percentages. Conversion of a good intron into a bad intron (**B**) or bad intron into a good intron (**C**) through change of hexamers. mCHERRY fluorescence is shown along with the RT-PCR amplification. The experiments were repeated twice in two independent experiments, which yielded similar results.

ways[8]: there can be sequence variation in a *trans*-acting factor modifying its function, or there can be a change in the potential binding site of a *trans*-acting factor. Both instances will result in splicing variation. While the former is likely to have an impact on multiple genes, the latter allows specific changes restricted to individual genes/splice-sites, generating variability which can be beneficial in evolution[73]. Our findings support the view that *cis*-regulatory changes are more common than *trans* regulatory sequence variation. We, however, acknowledge that due to our highly stringent analysis (i.e., having higher thresholds for peak calling and primarily looking at the top or closest associated SNP only), we could potentially underestimate the impact of *trans* variation, particularly if the effect size is relatively small. In addition, it is conceivable that by restricting our analysis to the highest associated SNPs, we could be missing out on factors that may be in LD with these SNPs. Nevertheless, confirming the causality of these *trans* associations would require further experiments.

We have presented a universal core logic that explains splice-site choice in eukaryotes based on hexamer rankings. Our findings that: (1). Hexamer prevalence and proportions of the used hexamers at splice-sites is correlated with splice-site strength rather than genomic prevalence; (2). Hexamer ranks explain splice-site choices; and (3). Hexamer ranks correlate across diverse species, together arguing for evolutionary selection for hexamer sequences.

Perhaps unsurprisingly, there is a near-perfect correlation between sequence distances from U1 snRNA base-pairing site and the average strength of intronic hexamers at splice donor sites (Supplementary Fig. 18B); the same intronic hexamers whose ranking explains much of the splice-site choice at a population level. The donor site is typically bound by U1 snRNA and sequence variability may influence the binding kinetics of U1 snRNA and its associated proteins, with potential consequences to splice-site choice[1]. In addition to snRNA-RNA base-pairing, some effects could also be attributed to hexamers (e.g., GTCTTT, GTCTTA) being binding sites for proteins such as PTBP1[74], which may interfere with splicing when the sequences are at the 5' donor sites[75]. Consistent with this, we found the average strengths of these hexamers to be very low (Supplementary Data 12). On the other hand, we observed hexamer GTAACG, which is a known binding motif for DAZAP1[74], a protein known to promote splicing, has a high average strength. Thus, in addition to snRNA-RNA base pairing, RNA-binding proteins could influence the average strengths. At the acceptor sites, where RNA-protein interactions play a critical role in splice-site selection[1,19,76], we observed similar correlations between the sequence distances from the highest ranked hexamer and the average strengths (Supplementary Fig. 18C). This can reflect differential binding of proteins with RNA. These differential interactions, while requiring further validation, may underlie the mechanism by which hexamer sequences influence splice-site choice.

Typical GWAS analysis results in the identification of genomic regions associated with phenotypic variation. Over the past 15 years, with the explosion of GWAS studies, there have been many genetic variants of interest that have been catalogued to be of significance[77]. However, understanding the implications of these SNPs and the underlying mechanisms by which they act remains a huge challenge[78]. Even restricting our analysis to the highest associated SNPs from SpliSER-GWAS, more than 10% have been identified to be variants of interest for diverse phenotypes (Supplementary Data 15). This suggests splicing as a potential molecular mechanism that could be explored in the context of these phenotypes, with important implications for personalized medicine.

It is important to point out some of the limitations of our study. SpliSER does not have any mapping capabilities and thus all mapping-related issues are carried over in our analysis. For example, we noticed that when we used different aligners, we had visible differences in scatter plots (Supplementary Fig. 6) and in the number of splice-sites. To assess the impacts, we compared the combinations of aligners and

accessions in Arabidopsis (Supplementary Fig. 6). We observed clear effects of the aligners on our splice-site strength estimations. However, when the same aligners were used on multiple accessions, they grouped together and separated from the same set of accessions aligned with a different aligner (Supplementary Fig. 6). Based on this, we reason that our ability to map SSE variation by GWAS is unlikely to be affected, since for SpliSER-GWAS analysis for each species, one aligner was used. It is also essential to note that our ability to map by GWAS is influenced by allele frequencies and thus our analysis will clearly miss rare alleles affecting splicing. For our analysis here, we have mostly focused on a single tissue from three different species, primarily due to the sheer computational time associated with carrying out hundreds of thousands of GWAS. Thus, we cannot rule out that we may have missed some patterns, which may become visible only when a comprehensive analysis of all tissues or cell types is carried out in the future.

It is worth noting that hexamer rankings, while providing valuable information, are not to be considered as an alternative to predictive tools. Hexamer rankings group all splice-sites into 256 categories and thus will never have the same predictive power as tools that use larger regions of sequences or contexts. In addition, these rankings can be developed in a tissue/condition/sex-specific manner, as opposed to sequence-based predictions. Thus, they provide a first and quick way to functionally assess the impacts across diverse organisms.

In summary, we have presented a framework that explains splice-site choice across eukaryotic organisms. Context-specificity (such as cell type-, tissue-, condition-specific splicing changes), perhaps mediated via context-specific expression of RNA-binding proteins, acts on top of this core framework to confer splicing outcomes. We have shown that the core determinant of splicing across diverse species is intronic hexamer sequences, and that these are among the major sources of splicing variability at a population level. Our catalogue of associations in Arabidopsis, Drosophila and Humans can be further exploited for functional analysis of pathways of growth, development, or environmental response. The hexamer rankings that we have presented can be used as a starting point for engineering specific splicing outcomes through gene editing techniques for desirable phenotypes of agricultural and/or medical relevance across eukaryotes.

## Methods

### Ethics Statement

This study did not involve any human participants but utilised human bulk RNA-seq data produced by the Genotype-Tissue Expression (GTEx) project, v8 release (dbGaP accession phs000424.v8; GTEx Portal, https://gtexportal.org/home/datasets). The datasets were provided access under the GTEx General Research Use (GRU)- Data Use Agreement upon review by NIH for this project (#30512 – Genome Wide Analysis of Splice-site choice). The data availability statement provides the links and the processes for obtaining the data. The use of data does not require approval from the human ethics committee. All data was used as per the requirements of the Data Use Agreement of NIH. Drosophila used in the study were maintained in accordance with institutional guidelines for the care and use of invertebrate animals in research. As Drosophila is not subject to the same ethical regulations as vertebrate animals, formal ethical approval was not required. Nevertheless, all efforts were made to minimize suffering and to handle the flies responsibly and humanly throughout the study. Please note the specific experiments (e.g., RNA extractions) were not presented in this manuscript, though we maintained the lines reported in the Drosophila GWAS data. Both Arabidopsis plant work and the work on human cell lines do not require formal ethics approval.

### DNA/RNA analyses

DNA and RNA extractions from plants and HEK293T cells (ATCC, Catalogue Number -CRL-3216) were carried out as described

previously[79,80]. Briefly, plant DNA was extracted using a modified CTAB protocol[79,80]. RNA extractions were carried out using Trizol (Invitrogen) as per manufacturer's instructions. Plants were grown under long day conditions (16 hr light and 8 hr dark) for expression analysis[79]. HEK293T cells were cultured as per standard procedures in DMEM media with 10% FBS and 2 mM glutamine. To analyze splicing, 1μg total RNA was converted into cDNA and the splicing patterns were analyzed using primers listed in Supplementary Table 5. All standard molecular biology works were done using standard protocols[79]. All constructs were sequence verified before their downstream uses. RT-PCR products were gel-purified and sequenced to confirm specific splice junctions.

### RNA-seq data alignment for processing

The Arabidopsis 1001 genomes RNA-seq data were aligned with TopHat2 and splice junctions were identified with Regtools[49]. Briefly, 6854 RNA-seq samples were downloaded, representing 728 accessions (PRJNA319904)[57]. Reads were aligned to the TAIR10 genome using TopHat2 (v2.1.1; parameters --minIntronLength 20, --maxIntronLength 6000, -p 6)[81]. For Drosophila, transcriptomic data were obtained from the Drosophila Genetics Reference Panel – DGRP2[59]. The 957 DGRP2 RNA-seq samples, representing 200 genotypes (both males and females each), were downloaded, and the data were aligned using STAR version 2.7[82] to the BDGP6 reference genome (parameters --outFilterMultimapNmax− --alignSJoverhangMin− --alignIntronMin 20 −alignIntronMax 150− --outSAMtype BAM SortedByCoordinate). Aligned BAM files were indexed with Samtools version 1.12. Regtools was used to generate gap junction files (parameters-regtools junctions extract -a 6 -m 20 -M 15000 -s 0). For the human data, 372 BAM files representing human heart atrial tissue generated by the GTEx project[60] were downloaded. A BED file also known as a gap junction file, containing a catalogue of splice junctions detected in the alignment was generated with Regtools junction extract (parameters -m 20 -M 16000 -a 6 -s 0 -o ${SAMPLE}.bed {SAMPLE}.bam).

### Quantification of splice-site strength/usage estimates (SSE)

The resulting BAM and splice junction BED files were processed with SpliSER v0.1.8[49]. We filtered sites that had at least 10 reads crossing the splice-site in at least three replicates, in at least 100 accessions and taken them for further analysis (Table 1). Fly data was split into female SSE (fSSE) and male SSE (mSSE) datasets. For each splice-site, SSE was averaged across samples of the same genotype and sex if at least two replicates passed the previous filtering steps.

Variability in splice-site usage between individuals was evaluated in two different ways. First, the variance of the distribution was calculated and individuals who fell in the upper quartile were taken for further consideration. Second, we computed the range in splice-site usage among the individuals in each of the species. Typically, there was a correlation between the two measures. Broad-sense heritability ($H^2$) was calculated from the averaged SSE phenotypic values as the proportion of total variance attributed to the variance between the DGRP/ 1001 Genome project accessions, using one-way ANOVA with the genotypes as a factor and SSE as a response. An earlier analysis with all sites in a set of flowering time genes suggested that high variability and heritability capture most of the mappable variation[49]. Therefore, splice-sites were ranked by their $H^2$ and total variance in SSE, and sites in the top quartile for both $H^2$ and total variance were taken for GWAS mapping. For the human heart data, all sites that were in the upper quartile of variance were taken for analysis since heritability could not be calculated due to the absence of replicates.

### Multi-tissue analysis of splicing in different species

RNA-Seq datasets for Chicken (*Gallus gallus*), Frog (*Xenopus tropicalis*), and Mouse (*Mus musculus*) were obtained from the publicly available NCBI BioProject repository under accession number PRJNA176589[9].

Human (*Homo sapiens*) RNA-Seq data were sourced from the Genotype-Tissue Expression (GTEx) Consortium[46], with a randomly selected individual's dataset. Splice-site strength was quantified using SpliSER (Version 0.1.8). Orthologous regions across species were established using the Lift-Over tool[83,84] (Galaxy platform) and regions with 1:1 overlap were taken further for comparative analysis as described by Barbosa-Morais et al.[9]. Differentially spliced genes were identified by constructing a 95% confidence interval around the regression line of SSE values between the two species or organs. Sites with SSE values falling outside this confidence interval were classified as differentially spliced.

### GWAS and peak-calling from Manhattan plots

We ran GWAS for each site using GEMMA v0.98.3[85], using genotypic data from the 1001 Genome project/DGRP project or the GTEx project. A distance square matrix kinship file was generated using the VCFs and then utilized in the GWAS analysis. Only the variants that had a minor allele frequency greater than 5% were considered for the association study. Initially, we manually called the peaks for Arabidopsis and then automated the process through benchmarking with Manhattan Harvester[86], with a minimum peak SNP count of 50. To account for the global noise in Manhattan plots in this approach, we additionally required that the −log10(pValue) of the top SNP of a peak must lie 1.33 times higher than the average of the top 5 SNPs in the plot (with none of the 5 being within 750 kb of each other) for the peak to be included. From each of the called peaks, a single SNP with the most significant p-value was selected as the top SNP. In cases where multiple SNPs had the maximal association significance, the one closest to the splice-site was chosen as the top SNP and selected for downstream analyses. All selected associations of an SNP and a splice-site were compiled into a comprehensive SNP table (Supplementary Data 1–3). The distance of each top SNP from the associated splice-site was calculated and normalised to the splice-site position and strand, with position zero being the first base in the intron for the donor site ("G" of "GT"), and last base of the intron for acceptor sites ("G" of "AG"). Associations were characterised as "*cis*" if the top SNP was within 1 Mb from the associated splice-site; all other associations were labelled as *trans*. Allelic change in SSE (ΔSSE) was calculated as the average major allele SSE subtracted from the average minor allele SSE. Percentage Variance Explained (PVE) of SSE by each SNP was calculated as per Shim et al.[87].

### Assessing the presence of motifs around Top SNPs

We extracted a +/-3-bp window sequence around the top SNP for associations and examined the presence of RNA binding motifs using sequence search option of ATtRACT database[74] (parameters: minimum motif length = 4, maximum motif length = 8). To investigate whether the allelic variation influences RNA binding, we compared the RNA binding motifs associated with both the major and minor alleles at the top SNP. For control, we performed the same analysis on a set of randomly selected SNPs or randomly generated sequences and conducted the same motif search analysis.

### Single nucleotide effects on splicing variation

We calculated the proportion of variance attributed to each position around the splice-site as the ratio between within-group variance and between group variance using this formula.

$$\text{Proportion of variance} = \frac{\text{TSS} - \text{RSS}}{\text{TSS}}$$

where TSS is the sum of squares of Splice-site Strength Estimates (SSEs) across all splice-sites, and RSS is the sum of the sum of squares of SSEs of sites with each nucleotide at the given position. We calculated this for all splice-sites in Arabidopsis, Drosophila and Humans, taking the information from multiple individuals that were used in the GWAS analysis.

## Second-order effects of nucleotides around splice-sites

For each species in the GWAS dataset we sampled down to approx. 2 million splice-site/sequence combinations. We investigated the sequences -7 upstream until +3 downstream of Acceptor Sites, and -3 upstream until +7 downstream of Donor sites. Preliminary analysis with two-way-ANOVA suggested that all positions around the splice-site were significantly associated with changes in splice-site usage, and that there were significant interaction effects of all nucleotides at all positions. To capture second-order effects (pairwise interactions of nucleotides which differ from a naive additive model), we calculated the mean splice-site usage of each splice-site with either nucleotide 1 (e.g., Adenine at position -7) and nucleotide 2 (e.g., Guanine at position -6; together forming an AG upstream of the splice-site). We subtracted the mean usage of sites with each nucleotide from the grand mean, and added the differences together to form an naive expectation of their interaction if it were additive (e.g., $-7A = -0.1$, $-6G = -0.15$, so we expect sites with $-7/-6$ AG to have an average usage 0.25 below the mean; capped such that splice-site usage could not exceed 0 or 1). We then looked at the difference between this expectation and our actual observations. We plotted all these differences in sorted order and identified outliers as those sitting beyond the elbows of the resulting plot (Supplementary Figs. 9–12). Elbow points were calculated as the points furthest from the lines drawn between the mid-point and either extreme of the plot, to which we added a further conservative buffer of 0.1 (or 10% change in splice-site usage beyond the elbow threshold). The pairwise interactions values are given in Supplementary Data 6, 7 and the heat maps of interactions for each species is shown in Supplementary Figs. 9–12.

## k-mer effects on splice-site choice

To identify an ideal k-mer length, we assessed the ability of the k-mer to explain splice-site choice in the following manner. We considered three aspects. First, we checked which length k-mers are present in the most possible sequence combinations at splice-sites. For example, a tetramer (e.g., GTNN) has 16 possible sequence combinations and all 16 could be found at the splice-sites. However, an octamer (GT $[N]_6$) has 4096 sequence combinations, but only some of these could be found at splice-sites. Therefore, a substantial number of octamers are not comparable across species. Second, we assessed the ability of k-mers to differentiate GT/AGs around splice-sites. The longer the k-mer, the more likely it is to uniquely differentiate a splice-site among competing sites. Third, we grouped all possible splice-sites into distinct k-mer groups, used their splice-site strengths to calculate the average strength of that k-mer and generated k-mer rankings, we scanned +/−100 or 200 bp from the splice-site for all GTs for donors and AGs for acceptors. We then extracted the k-mers surrounding the GT/AG in the window, taking into account of the existing mutations using corresponding VCF files for Arabidopsis, Drosophila and Humans and compared their average splice-site strength to generate k-mer ranks. The number of occasions when the splice-site had the strongest and unique k-mer among the k-mer for all GT or AG in that window were noted. This number is over the total analyzed sites to obtain a success rate. We then adjusted this score based on the percentage of possible k-mers that are present in splice-sites to obtain the percentage of splice-site choices explained by the k-mer (Table 2) for all three species. We excluded splice-sites that are within 100/200 bp of each other, eliminating the competition between detected sites for this analysis.

## Identifying the best nucleotides for each position that promote/suppress splicing

From the SNP table, we computed the distances of the highest and closest associated SNPs from their corresponding splice-sites. The table was filtered to only include splice-sites for which the highest associated SNP fell within 100 bp from the splice-site. We then trimmed down the list so that each associated SNP was giving information only for a single site by taking the closest splice-site. This resulted in a unique set of associated SNPs, each of which gave information about one unique splice-site. Subsequently, the list was separated into donor and acceptor sites, and the data were processed to produce the distribution graphs for each species. The data from all three species were combined and for each association,n the splice-promoting nucleotide and the splice-reducing nucleotides were deciphered as the most frequent nucleotide seen for greater or lesser usage of the splice-site. To synthesize the best and worst introns, we carried out this analysis on acceptors from -50 to 0 and from 0 to 50 for donors and stuck them together as shown in Supplementary Data 9. We included GT and AG and a branch point adenosine in all constructs in this analysis. The worst construct was then modified to include either a polypyrimidine tract (PPT) or a PPT and a consensus branch point or a PPT, a consensus branch point, and strong hexamers to assess the importance of hexamers. For all splice-sites across the genome, we computed the average SSE for each of the 4 possible nucleotides at each position relative to the splice-sites and plotted the means as a violin plot and considered the nucleotide with the highest mean as the best one to promote splicing for that position.

## mCHERRY reporter construct analysis

To assess the splicing impacts experimentally, we interrupted the mCHERRY ORF of pGH044 (Addgene Plasmid #85412, RRID: Addgene_85412). We scanned mCHERRY for the "AGGT" stretch and placed the intron between the AG and GT. We synthesized the intron with a part of mCHERRY through commercial suppliers (IDT-Australia) and then used restriction cloning to replace this cassette containing the intron within the plasmid pGH044. All constructs generated are listed in Supplementary Data 9. Sequence-verified constructs were transfected into HEK293T cells using Lipofectamine™ 3000 (Invitrogen). Cells were visualized under a fluorescence microscope. RNA from transfected cells was extracted for RT-PCR analysis.

## Hexamer analysis on diverse species

We downloaded the raw RNA-seq data (.fastq) of different species from SRA using SRAtoolkit. The RNA-seq reads were aligned to their corresponding genome annotations from SRA using STAR. Subsequent.bam files were then indexed and regtools was used to generate.bed files, detailing the splice junction information. Then, SpliSER was used to calculate splice-site usage. SpliSER's output file was used as input for a custom-made Python script that identified hexamers for each splice-site and calculated the average Splice-site Strength Estimate (SSE) for each hexamer. For Arabidopsis, Drosophila and human data, the entire dataset containing all individual samples (e.g., 200 genotypes in Drosophila) were used to generate obtain the hexamer for all sites, which further aided in obtaining a robust estimation of the SSE or count. For all other samples, single RNA-seq data was used to obtain the SSE or count ranks. These ranks were then used to assess the percentage of splice-site choices explained by the hexamers in each of these species. All the data from diverse species is reported in Supplementary Table 4 and Supplementary Data 12, 13. This data was used to obtain pairwise correlations between species through custom-made python scripts. The results are reported in Supplementary Data 14. To assess utilisation/nonutilisation of hexamer sequences, RNA-Seq data from a random individual was used. We analyzed the frequency with which each hexamer was utilized as a splice-site. For each hexamer, we determined its occurrence within protein-coding genes, recording both the count of hexamer usage as a splice-site and the corresponding SSE values across all individuals. A correlation analysis was performed, and scatter plots were generated to explore the relationship between hexamer presence and SSE. All analyses were conducted using custom Python scripts. To analyze species-wide correlations, we computed the *p-distance* using MEGA

software[88]. After grouping hexamers by *p-distance*, we computed the average SSE for each group by averaging the SSE of all hexamers within the group.

## Analysis of the Rosenberg et al.[30] minigene constructs data

We used the RNA-seq data generated by Rosenberg et al.[30], which was downloaded from NCBI in FASTQ format. As the reads were multiplexed, we used fastq-multx (https://github.com/brwnj/fastq-multx) to allocate sequences to different minigene constructs based on the barcodes. The de-multiplexed reads were aligned to the corresponding reference sequence using STAR. These aligned reads were processed with SpliSER (v1.8) and SSE values were obtained for all the detected splice-sites from all 265,137 constructs. Custom-made Python scripts were used in downstream analysis, which involved extracting the hexamer sequence information from the reference sequence and identifying all possible competing pairs within each construct. Subsequently, we calculated the average SSE value for all hexamers in each competing pair and determined the proportion of winning hexamers based on this average SSE value.

## Reporting summary

Further information on research design is available in the Nature Portfolio Reporting Summary linked to this article.

## Data availability

All data are presented in the supplementary tables and in the manuscript. The sequence data used are publicly available from the 1001 Genomes Project (Arabidopsis) with the accession number GSE80744 and the Drosophila Genotype Reference Panel with the accession number GSE67505. The human bulk RNA-seq data from heart atrial tissue and used in this study is available (with restrictions) from the GTEx portal and from dbGaP [https://www.ncbi.nlm.nih.gov/projects/gap/cgi-bin/study.cgi?study_id=phs000424.v8.p2]. The details on how to access the protected access data and the instructions on how to download the data are available on https://www.gtexportal.org/home/protectedDataAccess and https://support.terra.bio/hc/en-us/articles/4402326091675-Accessing-GTEx-TARGET-TCGA-data. The original data used in Figures S19 and S20 were generated by Barbosa et al., Science, 2012[9] and are available under the accession numbers GSE41338 and GSE30352. The Rosenberg et al., 2015[30] data used in Fig. 4 are available under accession GSE74070. For the multi-species rank-correlation analysis (Fig. S18), RNA-seq datasets were obtained from publicly available studies. The corresponding SRA run accessions (SRR) are listed in Source Data (Fig. S18_AccessionNumbers). To assess the effect of aligner choice (Fig. S6), we downloaded RNA-seq data from the Arabidopsis 1001 Genomes Project (SRA accessions: SRR3462994, SRR3462995, SRR3460130, SRR461135, SRR461134). Source data are provided with this paper.

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

## Acknowledgements

We thank the GTEx Consortium, 1001 Genomes Project and the Drosophila Genotype Reference Panel project and the researchers who made the transcriptome datasets available for scientific community use, which allowed us to undertake a cross-species approach. We thank Elaine Zhang, Hongyu Zheng, Shuyu Yan, Dhruv Sheth, Kai We Tan, Tenghao Zheng and Matthew Parker for their help and discussions and the members of the SKB/GATC labs for their critical comments on the manuscript. CID, JG, and JC are supported by an Australian Government Research Training Programme. MG is supported by an Indonesian Government Education Scholarship (Ref No. 202101030416). JR is supported by a Victoria Cancer Agency fellowship (Grant Number: MCRF20035). This work is supported by Australian Research Council Discovery Project DP190101479 (SB), Australian Research Council Future Fellowship (FT190100403) (SS), National Health and Medical Research Council Ideas Grant APP1182090 (SB & SS).

## Author contributions

Conceptualization: SB; Methodology: C.I.D., S.P., A.B., J.D.G.G., A.C., J.R., M.D.A., P.P.D., Y.L.G., A.F.L., S.S., D.P. and S.B.; Data curation: C.I.D., S.P., A.B., J.D.G.G., A.C. and S.B.; Software: C.I.D., S.P., A.B., A.C., A.F.L., D.P. and S.B.; Formal Analysis: C.I.D., S.P., A.B., J.D.G.G., A.C., S.M., J.C., M.G., R.D.S., R.B., S.S. and S.B.; Investigation: C.I.D., J.D.G.G., J.C., M.G., R.D.S., R.B., S.S. and S.B.; Writing – Original Draft: C.I.D., S.P., A.B. and S.B.; Writing-Review and Editing: C.I.D., A.C., M.G., S.P., A.F.L., Y.L.G., S.S., M.D.A., P.P.D., R.B. and S.B.; Visualization: C.I.D., S.P., A.B., A.C., J.C. and S.B.; Supervision: J.R., Y.L.G., A.F.L., R.B., S.S., D.P. and S.B.; Project Administration: SB; Funding acquisition: S.S. and S.B.

## Competing interests

The authors declare no competing interests.

## Additional information

[1]School of Biological Sciences, Monash University, Clayton Campus, Melbourne, VIC, Australia. [2]Department of Biochemistry and Molecular Biology and Cancer Program, Biomedicine Discovery Institute, Monash University, Clayton Campus, Melbourne, VIC, Australia. [3]Department of Medicine and Surgery, LUM University, Casamassima, Italy. [4]Gastrointestinal Genetics Lab, CIC bioGUNE – BRTA, Derio, Spain. [5]Ikerbasque, Basque foundation for Science, Bilbao, Spain. [6]Department of Anatomy and Developmental Biology, Monash University, Clayton, Melbourne, VIC, Australia. [7]Development and Stem Cells Program, Monash Biomedicine Discovery Institute, Clayton, Melbourne, VIC, Australia. [8]State Key Laboratory of Plant Diversity and Speciality Crops/State Laboratory of Systematic and Evolutionary Botany, Institute of Botany, Chinese Academy of Sciences, Beijing, China. [9]School of Biosciences, University of Melbourne, Parkville, VIC, Australia. [10]e-Research Centre, Monash University, Clayton Campus, Melbourne, VIC, Australia. [11]Present address: Max-Planck Institute for Plant Breeding Research, Cologne, Germany. [12]Present address: The Centre for Computational Biomedical Sciences, John Curtin School of Medical Research, Australian National University, Canberra, AT, Australia. [13]Present address: University of Rochester Medical Center, University of Rochester, New York, NY, USA. [14]These authors contributed equally: Craig I. Dent, Stefan Prodic, Aiswarya Balakrishnan. ✉e-mail: mb.suresh@monash.edu

