## [Transparent Peer Review file · Nature Communications]

A basic framework to explain splice-site choice in eukaryotes

Corresponding Author: Professor Sureshkumar Balasubramanian

Version 0:

Reviewer comments:

Reviewer #1

(Remarks to the Author)

In the manuscript, entitled 'A basic framework governing splice-site choice in eukaryotes', Dent et al. present a GWAS analysis with splice-site choice as the quantitative phenotype. The primary findings of the manuscript are (I) "hexamer prevalence at splice-sites is correlated with splice-site strength rather than genomic prevalence", (II) "Hexamer ranks explains splice-site choices", and (III) "Hexamer ranks correlate across species arguing for evolutionary selection." They also claim throughout the manuscript and abstract that (IV) "the most common genetically controlled variation in splicing is cis, and there are no major trans hotspots in any of the three species." The overarching claim of the manuscript is that splice-site hexamer strength is the basic framework of the splicing code in eukaryotes.

In my opinion, the analysis in the manuscript and the splice-site rankings as deliverables could be valuable to the scientific community. In particular, I found it intriguing that with all auxiliary cis-regulatory elements being unchanged, ablation of a dominant splice-site led to usage of the nearest strong alternative splice site.

However, I have concerns about whether the experimental design-- evaluating splicing in a single human tissue, and the stringent filtering applied-- is sufficient to support the broad conclusions drawn in the manuscript. This is a field that has been deeply studied over the past two decades, so I believe that the hexamer sequences themselves would benefit from greater evaluation in the context of other standard methods in the field. Without this, assessing the novelty of the findings is challenging, especially since splice-site sequence strength is already a well-recognized determinant of splice-site choice.

Below, I summarize several concerns and suggestions that, if addressed, could improve the manuscript.

Major:

1. The authors emphasize the role of core splice-site strength as the primary sequence determinant of a eukaryotic 'splicing code'. Yet, the decision to evaluate splicing in a single tissue may not adequately support the broad conclusions about these sequences as a 'splicing code' drawn in the study. For example:

(a) Previous multi-tissue GWAS investigations, such as Garrido-Martin et al. (PMID: 33526779), have suggested that functional variation in splicing across tissues occurs frequently through trans-factor regulation, reporting that many more variants modify the binding sites of RNA-binding proteins, than core splice-sites. This is not consistent with the major claims made here (conclusion IV).

(b) Barbosa-Morais et al. (PMID: 23258890) show that tissue-specific splicing has evolved more rapidly in vertebrates than gene expression has, underscoring a need for a multi-tissue approach to fully understand splicing mechanisms in the context of evolution. They also suggest that the 'splicing code' is tissue-specific and often driven by exonic and intronic splicing enhancer/silencer elements (binding sites of RNA-binding proteins), rather than core splice-site strength.

(c) Significant evidence implicates trans-factors in splicing regulation in disease. For instance, mutations in splicing factors such as SF3B1 and U2AF1 are recurrently found in cancer, and mutations in exonic splicing enhancers/silencers are linked

to many inherited diseases (PMID: 24456648). I'm curious if the authors have thoughts on how their model fits into such prior literature?

2. The manuscript would benefit from a comparison of their hexamer rankings with long-established splice-site scoring methods, such as the maximum entropy model by Yeo & Burge (PMID: 15285897). Such a direct comparison could help validate the findings novelty and determine whether the data presented here provides additional insights or improvements over standard methods.

3. Building on the previous point, I wonder how much better or worse the author's hexamer framework is able to explain splice-site choice, than the more sophisticated deep learning based methods they cite (eg. SpliceAI, Pangolin, SpliceVault, SPLAM). If the authors were able obtain comparable performance using their nearest highly ranked hexamer-based predictions versus these state-of-the-art models, this would strengthen the author's claims.

4. While the manuscript documents evolutionary conservation of hexamer ranks across species, it might be informative to evaluate this in the context of rapid splicing evolution observed in vertebrates, as noted by Merkin et al. (PMID: 23258891) and Barbosa-Morais et al. (PMID: 23258890). An analysis of evolutionary trends documented in these papers in the context of splice-site hexamer strengths presented here could strengthen conclusion III.

5. The authors assert that hexamer strength correlates with splice-site strength over genomic prevalence (conclusion I), but I'm concerned this oversimplifies the broader cis-regulatory context. Despite lower frequency in gene bodies, strong (but non-functional) splice-site hexamers may still vastly outnumber actual splice sites due to the extensive size of human gene bodies, which are often tens to hundreds of thousands of nucleotides long. Jaganathan et al. (PMID: 30661751) provide clear examples of this. Such data suggest that additional cis-regulatory elements beyond splice-site strength are crucial for splicing decisions, challenging the simplicity of the authors' framework. A deeper analysis of the absolute occurrence of utilized and un-utilized splice-site hexamers in gene bodies could be insightful here.

6. The manuscript asserts that the approach of using the author's software (SpliSER) to quantify individual splice-sites offers advantages over other cited methods. However, it is unclear to me how this differs significantly from other tools that do link splice-sites directly to percent-spliced-in (PSI) values, such as MAJIQ (PMID: 26829591). My understanding was that local-splicing-variation (LSV) is basically quantifying individual splice-site usage? Perhaps a comparative analysis using other splicing quantifications, might clarify the benefits or limitations of different quantification methods?

7. The manuscript focuses on mutations in core splice-site sequences, including the obligate GU-AG dinucleotides utilized by the major spliceosome. However, it is well-established that mutations in these obligate dinucleotides ablate splicing. Therefore such examples do not significantly advance our understanding of splicing regulation. Maybe I missed it, but what might be of greater novelty and value could be characterizing the more subtle and non-obvious variants still affecting splice-site strength, eg. What percentage of strong or weak splicing-sensitive variants are within or outside of these regions? What are the differences in effect levels of variants in the core splice-site dinucleotides vs other positions in splice-site hexamers, and compared to the upstream/downstream cis-regulatory regions? Along these lines, if you reduce the stringency of your filtering down from the top quartile of variance, does this distribution change?

8. The claim of establishing a "basic framework governing splice-site choice in eukaryotes" appears overly broad to me, particularly given the complex and context-dependent nature of splicing regulation in higher eukaryotes. It might be prudent for the authors to discuss potential limitations with respect to tissue-specific splicing context and their framework's ability to generalize in the presence of other well documented regulatory mechanisms that they may not be able to adequately measure with their approach.

Given these limitations, particularly the single-tissue analysis and stringent filtering for only the most splicing-sensitive variants, it may also be appropriate to reframe the manuscript's claims to focus within this scope. This may help to avoid overgeneralization and provide a clearer perspective on the study's contributions within the larger field.

Minor:

1. It would be informative to quantify the number of cis- vs. trans- variants detected in Figure 2, rather than qualitatively. Also I think it would be beneficial to investigate, when there are trans- variants detected, whether they fall into genes encoding RNA binding proteins, or gene enhancer regions or other? Also, how frequently do cis- variants fall distal in the gene body within the 1Mb window (in other exons or introns)?

2. In Figure 3, a significant portion of human variants lie deeper within exons and introns than the core splice-site recognition machinery would bind (eg. U2AF 65/35, and U1 snRNP). It is thought that these regions contain exonic and intronic cis-regulatory splicing elements (binding sites for RNA-binding proteins). Did the authors consider doing an exonic hexamer analysis (eg. Fairbrother et al. PMID: 12114529) on the variants that are lost or gained here-- similar to the approach taken in Lim et al. (PMID: 21685335) or Sterne-Weiler et al. (PMID: 21750108)?

3. In the statement "A vast majority of introns contain a consensus GU at their 5' end (donor) and an AG at their 3' end (acceptor), with a small proportion of introns harbouring alternative motifs (5' GC or AT and 3' AC) 7-10." It might be helpful to add that the GU-AG are dinucleotides of the major spliceosome and that alternative dinucleotides AT-AC are attributed to U12-type introns processed by the minor spliceosome.

4. I generally also feel the introduction could be improved, particularly with respect to the roles of alternative splicing in regulating crucial developmental processes, and in disease. It could also benefit from an introduction of exonic and intronic splicing enhancers and silencers upstream or downstream of the core-splice sites.

Reviewer #2

(Remarks to the Author)

In earlier work, these authors introduced SpliSER, a method for quantification of individual splice-site usage, and a score, SSE (Splice Site Strength Estimate), which ranges from 0 to 1 and is essentially the fraction of reads at the site that show its use as a splice site. Here, they use this measure in a genome-wide association study for all splice sites in three species (Arabidopsis thaliana, Drosophila melanogaster and humans). They observe that most of the variation in SSE is due to cis variation and use this large data set to infer the rules governing splice site usage in these three species.

The value of this paper lies in the comprehensive assessment of core splice site sequences with regard to something other than the properties of true splice sites. SpliceAI, for example, used a binary classification for training (each position in the genome was either a splice site or not a splice site; no measure of splice site strength was used. Here, the authors use data on two additional measures, SSE and splice-promoting allelic variation. However, rather than examine the difference between these sets (actual sequences vs. SSE vs. splice-promoting allelic variation), they focus on intronic hexamer sequences (GT[N]4 or [N]4AG), a subset of the core splice site signal that has been described in literally hundreds of prior papers. They don't argue that their hexamer ranking is a better predictor of splice site usage than (for example) MaxEnt (Yeo and Burge 2004; McCue and Burge 2024) or SpliceAI or any of several other tools, or compare the predictive power of hexamers to any of these predictors.

Another serious problem is that these authors introduce their own nomenclature for things that have well-established names. Most important here is the numbering of splice site nucleotides. In all papers published in the last 40 years, the nine nucleotides of the 5' splice site consensus MAG|GTRAGT are numbered -3, -2, -1, 1, 2, 3 etc.. Here, the same positions are numbered -3, -2, -1, 0, 1, 2 etc.. The 3' splice site has been similarly renumbered. I understand the desire to have the distance between positions be equal to the value of the difference in their position numbers. However, changing nomenclature after over 40 years of consistent literature will lead to confusion. For example, they talk about the importance of the "+4 guanine", which is referred to as position +5 in all previous work.

A lot of what they say here is a rediscovery of what is already known (e.g. the existence of an AG-exclusion zone, which they describe as a "non-additive pairwise interaction").

Major revisions:

1) The authors must report the trans-sQTLs that they have observed in each species. Although it is clear that most genetically associated splicing variation is in cis (as has been observed in all previous studies of sQTLs), some trans-sQTLs were observed (e.g. the region on Arabidopsis chromosome 5 to which several associations map, cited in Supplementary Figure S2). These appear as off-diagonal blue dots in the figures. Trans-sQTLs are of special interest and should be noted, even if they are rare. Are these known splicing factors? If so, which ones? They do mention one example (an unnamed GHMP kinase family protein in Arabidopsis).

2) The paper must be reorganized to foreground what has been learned by comparison of these results to prior work, which has been based solely on base frequency and the classification of sites as either true positives or true negatives.

3) The authors need to say a lot less about their hexamer ranking, unless they compare it to other measures (such as MaxEnt or SpliceAI and show it to be superior). The experimental work should be moved to supplemental data unless the authors can argue that their results differ from what would be predicted by existing models of splice site strength.

4) The numbering of nucleotides around the core splice site must be altered throughout the text to conform with standard nomenclature.

5) There needs to be a direct comparison with prior work on sQTLs in these three species, including a Venn diagram or upset plot showing which associations are shared between these studies. In particular, cite and compare Qi et al. 2022 <https://www.nature.com/articles/s41588-022-01154-4>

Garrido-Martin et al. 2021 <https://www.nature.com/articles/s41467-020-20578-2>

Khokar et al. 2019 <https://www.frontiersin.org/journals/plant-science/articles/10.3389/fpls.2019.01160/full>
Are there systematic differences?

6) There needs to be a discussion of the importance of LD in the interpretation of results.

6) The authors focus on nucleotides within 100 bp. of an affected splice site (e.g. Fig. 3 and Supplementary figure 5), and present megabase scale data. However, important genetic variation has significant long-range effects within a transcript (50 nt. to 20,000 nt.). How significant is genetically associated splicing variation at this distance?

7) The information shown in supplemental figures S10 through S14 must be made available as supplemental data tables.

Specific minor points:

8. the abstract should say what three species are examined.
9. "completing" in the legend to Fig. 1
10. In Fig. 2, PVE runs from 0 to 1 and is therefore presumably proportion rather than percentage.
11. There is no scale to the Fig. 1 A and B
12. The interactions shown by outliers in Fig. 3 E and F are presented in a very confusing way. Perhaps individual points could be labeled. This will take more space, but that would be justified.
13. Fig. 3. "highest associated SNPs" by what criterion? p value? difference in SSE?

Reviewer #3

(Remarks to the Author)

This study combines Genome-Wide Association Studies (GWAS) and a quantitative measure of splice-site usage termed Splice-site Strength Estimate from RNA-seq (SpliSER) to investigate RNA splice-site selection across three species: *Arabidopsis thaliana*, *Drosophila melanogaster*, and humans. The main conclusion is that splice-site choice is predominantly influenced by cis-regulatory elements, especially the hexamer sequences adjacent to the splice site. The key strength of the work is the innovative application of the SpliSER-GWAS method to examine how different genetic contexts affect splice-site selection. While GWAS has been previously applied to study splicing via splicing quantitative trait loci (sQTLs), this study is unique in treating splice-site usage itself as the phenotype.

However, there are two critical issues that, in my opinion, limit the biological significance of the study.

- 1) The absence of replicates in the human RNA-seq data is a serious limitation. In our experience, splicing decisions supported by RNA-seq datasets can be highly variable between replicates. Therefore, the higher variation in the usage of individual splice-sites observed in the human data may be significantly affected by technical noise. Including biological replicates would enhance the reliability and robustness of the findings.
- 2) The authors highlight their hexamer analysis as a novel biological observation. However, the significance of nucleotides surrounding splice sites is well established. Splice site sequences (including the hexamers in this study) have been used for over two decades to predict splice-site strength, as exemplified by tools such as MaxEntScan. The manuscript should acknowledge this existing knowledge and clarify how their approach offers new insights beyond established methods.
- 3) While the hexamer-based approach to predicting splice-site usage is informative, it does not account for the influence of splicing regulators. The authors begin their paper by stating that "changes in splicing can mediate phenotypic variation ranging from flowering time differences in plants to genetic diseases in humans." This phenotypic variation often involves the regulation of alternative splicing, which is not addressed in the study. The authors should analyze the impact of hexamers of varying strengths in contexts where annotated splicing enhancers or inhibitors are present. Such an analysis could provide deeper insights into the role of hexamers within the complex network of splicing regulatory sequences, enhancing our understanding of how these elements interact to govern splice-site selection. Additionally, the regulation of alternative splicing is highly tissue-specific, a factor that is ignored throughout the manuscript, particularly in the experiments involving the transfection of minigenes into HEK cells.

Additional comments:

- 4) The observation that trans variants are not prominently detected as hotspots in SpliSER-GWAS does not imply that these variants are irrelevant in splice-site selection. In fact, SpliSER-GWAS appears to be a promising tool for discerning the influence of trans variants. Given that the impact of cis-regional variants has already been extensively explored, both through functional studies and new machine learning algorithms, it is crucial to further investigate how trans variants might interact additively with local cis variants, including the previously mentioned hexamers, in influencing splice-site choices. This could provide a more comprehensive understanding of the genetic architecture underlying splice-site selection.
- 5) The claim that the results from this method are more interpretable than those obtained with SpliceAI might not be entirely accurate. While they may indeed be more reliable, the interpretability of these results remains limited. For example, the explanation that some hexamers are stronger than others at the 5' splice site due to their resemblance to the sequence recognized by U1 snRNA—stronger hexamers more closely mimic this sequence, whereas weaker ones diverge—is a reasonable observation. However, this alone does not suffice to deem the method as highly interpretable. Further analysis and explanation are needed to fully understand the nuances of how and why certain hexamers influence splice site selection more than others.
- 6) The supplementary tables provided at the end of the document, which detail the strength of each splice site, are indeed useful. However, creating an online tool that allows users to determine the hexamer strength of a splice site would be extremely valuable to the research community. Such a tool would not only facilitate easier access to this information but also enhance the practical application of the data in different research scenarios.
- 7) In this study, RNA-seq data from *Arabidopsis* were aligned using the TopHat2 aligner, while *Drosophila* and human data were processed with the STAR aligner. The choice of different aligners for different species raises questions, particularly since the central focus of the study is to understand what determines splice-site choice across different loci and species. The choice of aligner can significantly impact the mapping of spliced reads, which could influence the results. Ideally, it would be beneficial to standardize the use of aligners across all species studied to ensure consistency in data processing. Recognizing the substantial effort required to re-align the datasets, at a minimum, it would be necessary for the authors to demonstrate that both aligners perform comparably across all three organisms and across the full spectrum of splice-site strength estimates. This would help to validate the findings and confirm that the differing aligner choice does not skew the results.
- 8) "We filtered sites which had at least 10 reads crossing the splice-site in at least three replicates, in at least 100 accessions and taken them for further analysis." While this filtering criterion helps minimize false positives, it also likely reduces the detection of splice-sites that are lowly transcribed or infrequently used. This approach could significantly influence the results of downstream analyses. It would be informative to know whether the authors considered alternative thresholds.

Additionally, a rationale for setting the threshold at 10 reads would help clarify the basis for this specific choice, ensuring that it optimally balances sensitivity and specificity.

9) Page 7, paragraph 3: The authors should provide a detailed definition and context for what is meant by "GWAS peaks" in the study.

Version 1:

Reviewer comments:

Reviewer #1

(Remarks to the Author)

In revision, the authors of Dent et al., now entitled 'A basic framework to explain splice-site choice in eukaryotes', make improvements to their manuscript and provide some additional analyses which are helpful. However, some of the concerns raised by myself, and reviewers 2/3, remain.

In particular (as both myself and Reviewer 3 point out in the original review), despite the introduction & abstract highlighting regulatory examples of AS as the rationale for, and importance of, the work, the author's analysis still does not consider the tissue- context for alternative splicing (AS) signal thought to be relevant for many AS examples (as shown across decades of literature). Further, the authors still advertise the approach as a general framework to explain splicing decisions, despite this overarching blindspot (rather than a 'minimal basic framework' or other more attenuated statement).

The fact that the authors plan to focus on tissue-specific AS in another paper, does not exempt from reining in claims to only those that are justified given the data presented, or to provide additional analysis to support the broader claims of splicing generality.

While the revised version is better than the original draft, I feel that addressing the following major comments, would improve the manuscript:

1. In revision/rebuttal Dent et al. have made it clear that they are intending to quantify and explain the impact of genotype on splicing. However by focusing their analysis primarily on heart tissue (where muscle is one of the tissues with the lower frequency of AS reported; eg. PMID:15461793), they are in effect, studying how genotype can effect otherwise constitutive splicing, or a highly limited view of the effect of genotype on AS (much of which is regulated across tissues/cell-types). The addition of one more tissue does not greatly alleviate this limitation.

Moreover, the fact that the authors observe that most sQTLs reside outside of the core splicing motifs, and acknowledge that these likely alter RBP-binding sites, and given that many RBPs are differentially expressed across tissues, the effect of a genotype on splicing likely differs greatly depending on the tissue at hand. For example, it is plausible that the same SNP could enhance splicing in one tissue and repress it in another. This is an important concept that is completely lacking in this paper.

2. In the author's rebuttal re-analysis of the data from Barbosa-Morais et al., their interpretation is flawed/incomplete, as it does not include/consider that Barbosa-Morais et al. also report higher frequency of alternative splicing in primates and higher vertebrates, and greater frequency of tissue-specific splicing in brain tissue (nice overview here PMID:37993689). While there is also signal outside of the differentially spliced events (at given significance cutoff) that maintain species-specific clustering, that does not change the interpretation that tissue-specific splicing has more rapidly expanded/evolved in higher vertebrates (which is by definition, in a species-specific manner).

3. I'm not sure I agree with the author's assertion that there is a difference between wanting to 'explain splicing' and 'predict splicing'. If one can provide deterministic rules or categories that 'explain' splicing outcomes, then they are in essence providing a ruleset to predict splicing. If I am understanding the rebuttal correctly, the authors suggest that their framework benefits from being simple and interpretable, rather than accurate like AI methods. But does that not suggest that scoring splice-site hexamers is overly simplistic, and does not well capture the full cis-regulatory context that is essential for splicing determination?

Reviewer #2

(Remarks to the Author)

This revised version is acceptable with minor revisions (not requiring re-review). It is a significant and important contribution to the literature despite a tone that is dismissive of vast amounts of prior research.

Minor revisions:

1) The third sentence of the abstract ("However, how genetic variation influences splice site strength is largely unknown ...") strikes me as simply false, due to the vast literature on the impact of genetic variation on splicing. I think that what the authors mean is that splice site usage per se has not been previously used as it is here. The sentence has to be revised to

make it clear what is novel and what is not.

2) Similarly, "The rules that govern which GU/AG become splice site is still unclear." Should be changed to something like "... remain incompletely described."

3) pg. 15 - "optimal k-mer" should be optimal "intronic k-mer"

4) pg. 11 - "... trans associations (Table 2)" should be Table 1

5) pg. 13 - "these patterns are driven by not evolved neutrally" is agrammatical.

Reviewer #3

(Remarks to the Author)

The authors have substantially improved the clarity and depth of their analysis, included additional data that strengthens their conclusions, and more clearly positioned their findings within the context of existing literature. Importantly, they have also moderated some of the more speculative statements that were present in the original version.

While some of my original concerns have not been fully addressed, particularly in relation to potential artefacts due to lack of RNA-seq replicates and influence of splicing enhancers or inhibitors, I recognize that the revised manuscript represents a significant improvement overall. In my opinion, it will be of interest to the splicing community, as it provides a new approach to examine how different genetic contexts affect splice-site selection.

Version 2:

Reviewer comments:

Reviewer #1

(Remarks to the Author)

The authors have sufficiently addressed my major concerns.

Point by point Response Document

Thank you for your comments. We have done extensive revision of the manuscript with additional work including 7 new Supplementary Figures and 3 new Supplementary Tables along with changes in the text that addressed all the concerns of all reviewers. We have modified some sections of the manuscript in line with suggestions. Please find attached a detailed point-by-point response below.

Reviewer #1 (Remarks to the Author):

In the manuscript, entitled 'A basic framework governing splice-site choice in eukaryotes', Dent et al. present a GWAS analysis with splice-site choice as the quantitative phenotype. The primary findings of the manuscript are (I) "hexamer prevalence at splice-sites is correlated with splice-site strength rather than genomic prevalence", (II). "Hexamer ranks explains splice-site choices", and (III). "Hexamer ranks correlate across species arguing for evolutionary selection." They also claim throughout the manuscript and abstract that (IV) "the most common genetically controlled variation in splicing is cis, and there are no major trans hotspots in any of the three species." The overarching claim of the manuscript is that splice-site hexamer strength is the basic framework of the splicing code in eukaryotes.

In my opinion, the analysis in the manuscript and the splice-site rankings as deliverables could be valuable to the scientific community. In particular, I found it intriguing that with all auxiliary cis-regulatory elements being unchanged, ablation of a dominant splice-site led to usage of the nearest strong alternative splice site.

Thank you for the comments.

However, I have concerns about whether the experimental design-- evaluating splicing in a single human tissue, and the stringent filtering applied-- is sufficient to support the broad conclusions drawn in the manuscript.

We understand and agree with this point. In this manuscript, we are mostly focusing on genetic variation that drives splicing variability. Having said that, in the revision, we have addressed this concern by analysing additional tissues to a certain extent. We have refrained from a full analysis of all tissues for this manuscript as it is being done specifically exploring genetic basis of tissue-specific splicing in a different paper.

This is a field that has been deeply studied over the past two decades, so I believe that the hexamer sequences themselves would benefit from greater evaluation in the context of other standard methods in the field. Without this, assessing the novelty of the findings is challenging, especially since splice-site sequence strength is already a well-recognized determinant of splice-site choice.

We have provided more context in this extensive revision and addressed this concern directly with comparisons. We present now how our empirical estimations compare with predicted splice-site strengths and where the novelty of our work lies in this revision. We have also modified the title of the manuscript to reflect our position more accurately.

Below, I summarize several concerns and suggestions that, if addressed, could improve the manuscript.

Major:

1. The authors emphasize the role of core splice-site strength as the primary sequence determinant of a eukaryotic 'splicing code'. Yet, the decision to evaluate splicing in a single tissue may not adequately support the broad conclusions about these sequences as a 'splicing code' drawn in the study.

Our work is primarily cataloguing genetic variation associated with splicing differences by GWAS in distinct species. In addition, this is the first attempt to directly link the usage of specific splice-sites with genetic variation. This is different from previous sQTL analyses, as the phenotypes used for association are very different (more on this below). In particular, a single tissue is required to address genetically controlled splicing variation, which is the main question we want to address. We have indeed analysed additional tissues, and we have presented some aspects of this in the new Supplementary Figure 1. Please see our point-by-point response to all concerns below, including how we have addressed them in the revised manuscript.

For example:

(a) Previous multi-tissue GWAS investigations, such as Garrido-Martin et al. (PMID: 33526779), have suggested that functional variation in splicing across tissues occurs frequently through trans-factor regulation, reporting that many more variants modify the binding sites of RNA-binding proteins, than core splice-sites. This is not consistent with the major claims made here (conclusion IV).

Thank you for this comment. Our conclusion IV that “the most common genetically controlled variation in splicing is *cis*” is not in conflict with the observations of Garrido-Martin et al. Garrido-Martin states that many more variants modify binding sites of RNA-binding proteins, than core splice-sites. In fact, our analysis also shows that only few variants directly affect core splice sites [GT/AG] (1% (32Top SNP[TS]/53Closest SNP[CS]/5277) in Arabidopsis, 2% (117TS/155CS/7791) in Drosophila and 0.35% (35TS/35CS/9872) in humans). Even in the extended regions (hexamer region) there are only 6.74% (221TS/356CS/5277) in Arabidopsis, 6.21% (279TS/484CS/7791) in Drosophila and 0.89% (97TS/88CS/9872) in humans). Therefore, the majority of the specific variants we map, affect other regions (or have indirect effects on other sites, for example, mutations at donor sites also affect partner or competitor sites). Therefore, our findings are not inconsistent.

Perhaps, it is important to note that GWAS tells us about “regulation of variation in splicing” rather than “regulation of splicing”. While there may be many trans-regulators of splicing, our GWAS analysis suggests that allelic variation is less common in these regulators. Once again this is consistent with Garrido-Martin’s idea of variants modifying “trans-factor binding sites”. In fact, variants that modify the binding sites of RNA-binding proteins are also “cis” variants though they modify the

binding of “trans” factors. In other words, our conclusion that most splicing variation is “cis” regulatory is accurate, although obviously splicing can be regulated by trans-factors. Therefore, there is no inconsistency with our claims (conclusion IV).

We would also like to point out that the sQTL analysis by Garrido-Martin et al is not a genome-wide analysis but is restricted to variants present in the surrounding genomic context (analysing variants +/- 5-10Kb of the gene) trying to identify “cis” sQTLs only. Because their analysis is based variants in +/-5Kb region of the gene, in principle, their analysis simply does not aim to detect any “trans” regulatory variation, even if present. We have now presented a supplementary table and a figure, where we have explicitly highlighted the differences between published sQTL studies and our analysis (Supplementary Table 6, Supplementary Figure S3). Our study for the first time in the GTEx data tested everything by GWAS as opposed to a localised sQTL analysis from a few Kbp to 2Mb region around genes.

(b) Barbosa-Morais et al. (PMID: 23258890) show that tissue-specific splicing has evolved more rapidly in vertebrates than gene expression has, underscoring a need for a multi-tissue approach to fully understand splicing mechanisms in the context of evolution. They also suggest that the 'splicing code' is tissue-specific and often driven by exonic and intronic splicing enhancer/silencer elements (binding sites of RNA-binding proteins), rather than core splice-site strength.

We thank the reviewer for raising this important point, which gave us an opportunity to address this issue and clarify some aspects of these findings. While our findings are not inconsistent with the data, our interpretations of these findings differ as we explain below.

We agree that core splice-site strength cannot explain tissue-specific differences. Barbosa-Morais et al compared differentially expressed genes (DEG) and differentially spliced genes (DSG) based on alternatively spliced exons and showed that while DEGs cluster by tissue across species, DSGs cluster by species. Based on this observation, they concluded that tissue-specific splicing evolved more rapidly than gene expression. While this can be one interpretation of this observation, we believe this data, in fact suggests that gene expression is regulated mostly at the tissue level and, in contrast, splicing is regulated at the organismal level thus consistent within the species/lineage.

To assess these two alternative possibilities, we tested whether the clustering of sites that show tissue/species-specific usage differs from the clustering of sites that are not differentially spliced in tissues and/or species. If the pattern is primarily explained by tissue-specific splicing differences, one would expect the clustering to differ between sites that do not splice in a tissue-specific manner. We re-analysed the data from Barbosa-Morais et al running SpliSER on them, which provided information at multiple independent sites with increased statistical power. We then tested whether clustering differs if we analyse: a) the entire data set; b) sites that do not show differential splicing between species or tissues; c) sites that differ between species for the same tissue; or d) sites that differ between tissues for the same species. Irrespective of the split in the data, we observed clustering based on species, which suggests that this trend is not related to tissue-specific splicing but

rather associated with general species-level changes in splicing regulation. These data also suggest that the core of the splicing code does not differ between tissues, and that tissue-specificity in splicing is an attribute of the splicing code that acts on top of the basic framework. We have now presented this analysis as a Supplementary Figures S19 and S20 in the revised manuscript and addressed this in the manuscript.

(c) Significant evidence implicates trans-factors in splicing regulation in disease. For instance, mutations in splicing factors such as SF3B1 and U2AF1 are recurrently found in cancer, and mutations in exonic splicing enhancers/silencers are linked to many inherited diseases (PMID: 24456648). I'm curious if the authors have thoughts on how their model fits into such prior literature?

Our findings do not rule out a role of trans-factors in splicing regulation in disease. In fact, a simple analysis with our Top SNPs suggests that many of these will make a difference to the binding sites of RNA binding proteins. However, our ability to map mutations such as SF3B1 and U2AF1 is limited by the frequency of these alleles in our dataset. Our data suggests that there is no common allelic variation in splicing factors such as SF3B1 and U2AF1 in general population. So, we do not think our findings conflict with these observations.

2. The manuscript would benefit from a comparison of their hexamer rankings with long-established splice-site scoring methods, such as the maximum entropy model by Yeo & Burge (PMID: 15285897). Such a direct comparison could help validate the findings novelty and determine whether the data presented here provides additional insights or improvements over standard methods.

Thank you for this comment. We have addressed these in the following ways.

- 1) We now directly compared how predicted splice-site strengths by MaxEnt reflect empirical quantifications of splice-site strength by SpliSER. Empirical quantifications generally differ from MaxEnt predictions, and not reflective of empirical quantifications. This is now presented in Supplementary Figure S1.
- 2) MaxEnt splice-site score or other established splice-site scoring methods are sequence-based predictions and are not empirical quantifications, which would mean that they will not differ for example, between tissues, organs, sex or conditions. In contrast, SpliSER-computed SSE is an empirical quantification, and all these context-dependent differences can be captured differentiating SpliSER quantifications from sequence-based predictive scores (Supplementary Figure S1).
- 3) We have also compared the MaxEnt scores to Hexamer rankings and show that MaxEnt scores indeed can explain splice-site choices better than Hexamers. However, we also show that majority of the MaxEnt scores themselves are primarily driven by hexamer sequences and the MaxEnt rankings are highly correlated with hexamer rankings. Both these aspects are discussed in the text and as supplementary Figure S16.

4) We recognized upon retrospection we have not communicated our thoughts on hexamers clearly. We do not present hexamer rankings as a “splice-site scoring method” but rather as a minimal sequence feature that can be compared across species to explain variability in splice-site choices. We do not view hexamers as another predictive tool that is comparable to any tools that are currently available. Neither do we believe or claim that we have discovered and present the importance of sequences surrounding splice-sites, which has been indeed the subject of many previous papers. What we uncover and demonstrate is that there is a potential ranking order of all possible hexamer sequences, which we can compute based on empirical usage (which was simply not feasible earlier since there were no actual measures of usage), as opposed to distance matrix analysis as Position Weight Matrices (which are not based on empirical usage) and they do explain a substantial proportion of variation in splice-site usage. As such, this represents one of the simplest and basic features to explain splice-site choice, where the rankings can even reflect differences between genotypes, conditions, tissues etc. This allows comparing rankings between any of these combinations as well as across species. We have now revised the manuscript including title and abstract and hope that our nuanced position is now clearly reflected in the manuscript.

3. Building on the previous point, I wonder how much better, or worse the author's hexamer framework is able to explain splice-site choice, than the more sophisticated deep learning-based methods they cite (e.g. SpliceAI, Pangolin, SpliceVault, SPLAM). If the authors were able obtain comparable performance using their nearest highly ranked hexamer-based predictions versus these state-of-the-art models, this would strengthen the author's claims.

We agree that these sophisticated deep learning methods will be way better in predicting splice-sites from sequence, but the underlying reasonings for their predictions are unclear. In fact, some of these methods (Pangolin – Zeng & Li, *Genome Biology*, 2022; SpliceBERT – Chen et al, *Brief. in Bioinformatics*, 2024; SpliceTransformer – You et al, *Nat. Comm*, 2024), used SpliSER to empirically estimate splice-site usage and trained their models or tested their predictions using this dataset. Our approach here is not to develop a predictive tool. Hexamers classify splice-sites into small comparable categories/groups across species/genotypes/tissues to explain splicing choices across eukaryotes.

4. While the manuscript documents evolutionary conservation of hexamer ranks across species, it might be informative to evaluate this in the context of rapid splicing evolution observed in vertebrates, as noted by Merkin et al. (PMID: 23258891) and Barbosa-Morais et al. (PMID: 23258890). An analysis of evolutionary trends documented in these papers in the context of splice-site hexamer strengths presented here could strengthen conclusion III.

Thank you for this comment and suggestion. We have now presented hexamers in the context suggested in Supplementary Figures S19-S20, which further strengthens our conclusion that indeed hexamer ranks correlate across species as a function of evolutionary distance. We used the same data across tissues and show that the

hexamer ranking correlations between tissues from the same species is stronger than across species, supporting the same evolutionary trend already described in the paper.

5. The authors assert that hexamer strength correlates with splice-site strength over genomic prevalence (conclusion I), but I'm concerned this oversimplifies the broader cis-regulatory context. Despite lower frequency in gene bodies, strong (but non-functional) splice-site hexamers may still vastly outnumber actual splice sites due to the extensive size of human gene bodies, which are often tens to hundreds of thousands of nucleotides long. Jaganathan et al. (PMID: 30661751) provide clear examples of this. Such data suggest that additional cis-regulatory elements beyond splice-site strength are crucial for splicing decisions, challenging the simplicity of the authors' framework. A deeper analysis of the absolute occurrence of utilized and unutilized splice-site hexamers in gene bodies could be insightful here.

We agree that additional cis-regulatory elements beyond splice-site strength affect splicing decisions as considerable fraction of our associations map beyond splice sites. To address reviewer's query, we have compared the absolute occurrence of utilised and unutilized splice-site hexamers in gene bodies in all three species and present that data in the revision (Supplementary Table S14 and S15).

We found that:

- a) the number of hexamers at the splice-sites is correlated with their average SSE rather than their actual number in gene bodies ($R^2=0.7$ for donors and 0.64 for acceptors) [Supplementary Figure S17]
- b) The number of utilized hexamers is NOT correlated with the number of unutilised hexamers in gene bodies ($R^2=0.04$ for donors and 0.001 for acceptors).
- c) The average SSE of hexamers is NOT correlated with the number of unutilized hexamers ($R^2=0.001$ for both donors and acceptors)
- d) The fraction of utilised hexamers is correlated with the average SSE of the hexamer ($R^2=0.74$ for both donors and acceptors).

These findings are consistent with the idea that hexamer strength indeed correlate more with splice-site strength than their genomic prevalence. While the unused hexamers do outnumber used hexamers, their rankings based on their absolute numbers differentiates used hexamers, and the percentage of the used of hexamers also correlates with their average SSE. Taken together, these data argue for biological relevance of hexamers being key sequence features that could explain splice site choices. We however agree that our findings do not rule out the role of other cis-regulatory variants and our idea does not contradict the findings of Jagannathan et al, where in specific cases other sequences might play a critical role in splicing decisions.

6. The manuscript asserts that the approach of using the author's software (SpliSER) to quantify individual splice-sites offers advantages over other cited methods. However, it is unclear to me how this differs significantly from other tools that do link splice-sites directly to percent-spliced-in (PSI) values, such as MAJIQ (PMID: 26829591). My understanding was that local-splicing-variation (LSV) is basically quantifying individual splice-site usage? Perhaps a comparative analysis using other splicing quantifications, might clarify the benefits or limitations of different quantification methods?

We have presented these comparisons in our previous manuscript (PMID: 34017946), including a comparison with MAJIQ. Briefly, the key difference in our approach is that the SSE values are specific to individual donor and/or acceptor sites as opposed to quantification of splice junctions. In other words, SSE values can be obtained for every single base pair of the genome based on the reads that map to that base pair. Therefore, this is different from other methods although the logic is similar to PSI values. We would posit that SpliSER is complementary rather than better or worse. SpliSER quantifications could be directly used as phenotypes in GWAS, specifically for each splice donor or acceptor site, which is not feasible with MAJIQ for example.

7. The manuscript focuses on mutations in core splice-site sequences, including the obligate GU-AG dinucleotides utilized by the major spliceosome. However, it is well-established that mutations in these obligate dinucleotides ablate splicing. Therefore, such examples do not significantly advance our understanding of splicing regulation. Maybe I missed it, but what might be of greater novelty and value could be characterizing the more subtle and non-obvious variants still affecting splice-site strength, eg. What percentage of strong or weak splicing-sensitive variants are within or outside of these regions? What are the differences in effect levels of variants in the core splice-site dinucleotides vs other positions in splice-site hexamers, and compared to the upstream/downstream cis-regulatory regions? Along these lines, if you reduce the stringency of your filtering down from the top quartile of variance, does this distribution change?

We only used the GU-AG dinucleotides to assess whether SpliSER-GWAS approach can map the variation in splicing directly to the causal nucleotide. Indeed, the majority of the variants map outside core splice-site sequences. To assess whether there is a significant difference in effect sizes, we have now compared the SNP effects (differences in splice-site strength [dSSE]) based on their proximity to splice site (Supplementary Figure S7). SNPs if they are present in the same gene, had a higher impact if they are in the same intron/exon as that of splice site. However, when the SNPs are in different genes the effects were similar to SNPs that are present in the same exon/intron, which indicates that the distance of the SNP had limited impact, if we are able to map the splicing differences. We do not, however, think that relaxing the filtering threshold down from top quartile would affect distribution. This was tested previously (Dent et al, NAR Genomics and Bioinformatics, 2021) using a subset of genes to validate this upper quartile cut-off.

8. The claim of establishing a "basic framework governing splice-site choice in eukaryotes" appears overly broad to me, particularly given the complex and context-dependent nature of splicing regulation in higher eukaryotes. It might be prudent for the authors to discuss potential limitations with respect to tissue-specific splicing context and their framework's ability to generalize in the presence of other well documented regulatory mechanisms that they may not be able to adequately measure with their approach.

We have discussed the limitations of this work now in a separate paragraph in discussions. We however think that our framework is flexible enough to account for the differences and complexities arising from tissue /organ / cell type / conditional

specificities in splice-site choices. In other words, while the hexamer rankings remain mostly unchanged between tissues, suggestive of a general framework at a species-level, we have also mapped additional regions that confer tissue specificity. We have touched upon briefly on these topics in this manuscript, while a subsequent manuscript detailing a catalogue of tissue-specific motifs is currently in preparation.

Given these limitations, particularly the single-tissue analysis and stringent filtering for only the most splicing-sensitive variants, it may also be appropriate to reframe the manuscript's claims to focus within this scope. This may help to avoid overgeneralization and provide a clearer perspective on the study's contributions within the larger field.

Thanks for this suggestion. We agree that we have stringent filtering, but we are not filtering for large effects, but rather on association/mapping strength in GWAS, which is inherently limited by allelic frequency and variability in phenotypes, but not by splicing sensitivity. Our data clearly show that most sites explaining at least 20% difference in the usage of splice-site are in the upper quartile of variability in the splicing phenotypes. With the limitations of the work now explicitly stated, the manuscript now presents a clearer perspective on this contribution within the larger field.

Minor:

1. It would be informative to quantify the number of cis- vs. trans- variants detected in Figure 2, rather than qualitatively. Also, I think it would be beneficial to investigate, when there are trans- variants detected, whether they fall into genes encoding RNA binding proteins, or gene enhancer regions or other? Also, how frequently do cis-variants fall distal in the gene body within the 1Mb window (in other exons or introns)?

The information on the number of cis- and trans- variants is already provided in the Table 1. Our top SNPs (single SNP for each trans association) do not co-localise, which suggests that there are no genomic hot spots. However, it is possible that other SNPs in the GWAS peak and SNPs that are in LD with these can fall into diverse categories. However, we do not think it would be meaningful to draw conclusions on these on a global scale. We believe that each of these trans-regions requires individual analysis, which is beyond the scope of this manuscript.

With respect to the distances, we have now presented the effect based on whether the SNP is in the same intron/exon of the splice site, in the same gene or in different genes. Supplementary Figure S7. Of the 1564, 2005 and 2592 instances where the Top SNP was on the same gene in humans, Arabidopsis and Drosophila, respectively, 836 (53%), 1053(52.5%), 1025(39.5%) of these were not in the same exon/intron as the splice-site.

2. In Figure 3, a significant portion of human variants lie deeper within exons and introns than the core splice-site recognition machinery would bind (eg. U2AF 65/35, and U1 snRNP). It is thought that these regions contain exonic and intronic cis-regulatory splicing elements (binding sites for RNA-binding proteins). Did the authors consider doing an exonic hexamer analysis (eg. Fairbrother et al. PMID: 12114529)

on the variants that are lost or gained here-- similar to the approach taken in Lim et al. (PMID: 21685335) or Sterne-Weiler et al. (PMID: 21750108)?

Given that we can genetically map the variants that are associated with a change in splicing, we asked whether the most associated SNP falls within the binding site for an RNA binding protein and whether that binding is altered with the variants. Out of the 3949 unique SNPs that we have identified as Top SNPs, we found 3387 (86%) affect the binding site of an RBP. In addition, majority of these (>90%) are not among the core-splice sequences. However, there are some limitations with the suggested frequency-based approaches. Given the number of RBPs and small size of their binding motifs (5-6 bp), even random sequences have the potential to give an impression affecting RBP binding. We observed the same pattern on simulated data, which justified not focusing on this result in the absence of further experimental evidence.

3. In the statement "A vast majority of introns contain a consensus GU at their 5' end (donor) and an AG at their 3' end (acceptor), with a small proportion of introns harbouring alternative motifs (5' GC or AT and 3' AC) 7-10." It might be helpful to add that the GU-AG are dinucleotides of the major spliceosome, and that alternative dinucleotides AT-AC are attributed to U12-type introns processed by the minor spliceosome.

Modified and rephrased in the revision.

4. I generally also feel the introduction could be improved, particularly with respect to the roles of alternative splicing in regulating crucial developmental processes, and in disease. It could also benefit from an introduction of exonic and intronic splicing enhancers and silencers upstream or downstream of the core-splice sites.

We have modified the introduction with the suggested edits.

Reviewer #2 (Remarks to the Author):

In earlier work, these authors introduced SpliSER, a method for quantification of individual splice-site usage, and a score, SSE (Splice Site Strength Estimate), which ranges from 0 to 1 and is essentially the fraction of reads at the site that show its use as a splice site. Here, they use this measure in a genome-wide association study for all splice sites in three species (*Arabidopsis thaliana*, *Drosophila melanogaster* and humans). They observe that most of the variation in SSE is due to cis variation and use this large data set to infer the rules governing splice site usage in these three species.

The value of this paper lies in the comprehensive assessment of core splice site sequences with regard to something other than the properties of true splice sites. SpliceAI, for example, used a binary classification for training (each position in the genome was either a splice site or not a splice site; no measure of splice site strength was used. Here, the authors use data on two additional measures, SSE and splice-promoting allelic variation. However, rather than examine the difference between these sets (actual sequences vs. SSE vs. splice-promoting allelic variation), they focus on intronic hexamer sequences (GT[N]4 or [N]4AG), a subset of the core

splice site signal that has been described in literally hundreds of prior papers. They don't argue that their hexamer ranking is a better predictor of splice site usage than (for example) MaxEnt (Yeo and Burge 2004; McCue and Burge 2024) or SpliceAI or any of several other tools or compare the predictive power of hexamers to any of these predictors.

We apologies for not being effective in communicating our thoughts clearly. We have modified the manuscript to address this deficiency in this revision in the following ways.

First, we have now compared how measures like MaxEnt score compares with empirical quantifications early in the manuscript (Supplementary Figure S1). Second, we made it explicit in the manuscript that hexamer sequences are not to be considered as an alternate or competing approach with SpliceAI or MaxEnt. Many tools (Pangolin – Zeng & Li, *Genome Biology*, 2022; SpliceBERT – Chen et al, *Brief. in Bioinformatics*, 2024; SpliceTransformer – You et al, *Nat. Comm*, 2024) actually integrate SpliSER to improve the accuracy of splicing prediction models. Though we do not consider hexamers as predictors, hexamers also explain most of the MaxEnt scores (Supplementary Figure S16). Hexamers reduce hundreds of thousands of splice-sites into 256 donor & 256 acceptor classes and this dimensionality reduction means that they cannot be better predictors than the complete sequence information.

We present hexamers as reduced sequence features that can explain most of the variability in the data. We agree that several papers may have pointed to the importance of the hexamer region. As the reviewer points out that it is well known that MAG/GTRAGT is known to be the splice-site consensus, but the hexamer GTRAGT occurs only in 23% of the human splice sites (12,730,991 out of 56,508,411 sites in the GTEx data). However, how variability in these sequences is associated with variability in splice site strength is not something that has been explained in any of these papers. Here we have presented a rank order based on empirical estimations and that is the contribution of this paper.

Another serious problem is that these authors introduce their own nomenclature for things that have well-established names. Most important here is the numbering of splice site nucleotides. In all papers published in the last 40 years, the nine nucleotides of the 5' splice site consensus MAG|GTRAGT are numbered -3, -2, -1, 1, 2, 3 etc. Here, the same positions are numbered -3, -2, -1, 0, 1, 2 etc. The 3' splice site has been similarly renumbered. I understand the desire to have the distance between positions be equal to the value of the difference in their position numbers. However, changing nomenclature after over 40 years of consistent literature will lead to confusion. For example, they talk about the importance of the "+4 guanine", which is referred to as position +5 in all previous work.

We have now modified this as suggested. Wherever we refer to nucleotides surrounding the splice-sites, we have changed numbering as the reviewer suggested. Only in our distance calculations, we kept our system (e.g., In supplementary SNP tables showing associations and distances from the splice site "G").

A lot of what they say here is a rediscovery of what is already known (e.g. the existence of an AG-exclusion zone, which they describe as a "non-additive pairwise interaction").

We have mostly removed the sections on the discussions of pairwise interactions in this revision and only presented the actual data. We have also carefully revised the manuscript to ensure that we highlight novel aspects of our work clearly. Having said that, AG-Exclusion Zone is defined based on the suggested preference of the first AG after the branch point being used a splice acceptor site and thus not having any AG before the actual splice acceptor AG (Gooding et al, Genome Biology, 2006). However, it is also known in several species that there are NAGNAG sites that cause alternative splicing (Hinzpeter et al, PLoS Genetics, 2010; Bradley et al, PLoS Biology, 2012; Lida et al, MBE, 2008; Busch and Hertel, Genome Biology, 2012; Yan et al, FEBS Letters, 2015).

Major revisions:

1) The authors must report the trans-sQTLs that they have observed in each species. Although it is clear that most genetically associated splicing variation is in cis (as has been observed in all previous studies of sQTLs), some trans-sQTLs were observed (e.g. the region on Arabidopsis chromosome 5 to which several associations map, cited in Supplementary Figure S2). These appear as off-diagonal blue dots in the figures. Trans-sQTLs are of special interest and should be noted, even if they are rare. Are these known splicing factors? If so, which ones? They do mention one example (an unnamed GHMP kinase family protein in Arabidopsis).

We have reported trans associations as well in Supplementary Tables S2-S4. In the revised manuscript, we have included a separate paragraph on the "trans" associations, including a discussion on why we might need to be cautious in interpreting these trans associations. For example, the GHMP kinase trans association in Arabidopsis appear to have resulted due gene duplications that are seen only in some accessions.

We would however point out that our SpliSER-GWAS differs from earlier sQTLs in a few ways:

First, we reported only the most associated SNP for each peak and not all SNPs that show statistical significance as done in all previous sQTL analysis. In a direct comparison, there could be multiple (from 1-2 to several hundred) SNPs in a GWAS peak that would be technically considered as sQTLs in analysis.

Second, most of the previous sQTL studies are not unbiased genome-wide association studies but rather look for variants within short region only (5 to 10kb or up to 1 Mb region, Supplementary Table S6). Since, sQTL studies are only testing for *cis* effects, they simply cannot detect any trans sQTLs and thus do not quantify whether most of the associated variation is *cis* or *trans*.

Finally, SSE is a more accurate phenotype for mapping purposes and given mapping is a function of the accuracy of the phenotype, we believe our study has higher power and resolution than previous sQTL analysis to capture genetic variation affecting splice site choice. We have compared directly (Supplementary Table S6) our study with previously published studies and presented some example differences in a Supplementary Figure S3.

2) The paper must be reorganized to foreground what has been learned by comparison of these results to prior work, which has been based solely on base frequency and the classification of sites as either true positives or true negatives.

We have revised the manuscript to highlight the specific aspects that are novel and learnt from these studies compared to previous studies.

3) The authors need to say a lot less about their hexamer ranking, unless they compare it to other measures (such as MaxEnt or SpliceAI and show it to be superior). The experimental work should be moved to supplemental data unless the authors can argue that their results differ from what would be predicted by existing models of splice site strength.

Thank you. We have toned down the hexamer ranking section considerably. As we clarify in the text, AI models such as Pangolin, SpliceBERT and SpliceTransformer are trained/tested on empirical quantification provided by SpliSER (Zeng and Li, Genome Biology, 2022; Chen et al, Brief Bioinformatics, 2024; You et al, Nat. Comm, 2024), which we believe is the proper avenue for the use of our earlier work in splice site prediction. We do not rely on hexamer rankings as predictor, but rather a feature with reduced dimensionality, which provides a rationale for why a particular site may be able to compete out another splice-site, based on empirical computation of average intrinsic splice-site strength. Hexamer rankings allow us to compress and summarise hundreds of thousands of splice-site sequences into 256 donor and 256 acceptor categories.

Given this context, we disagree with the repositioning of our experimental work. We designed the specific mutations in the hexamer sequences, based on empirically estimated average hexamer strengths. In addition, we also have a specific case of a specific polymorphism that we identified as an associated SNP. Therefore, these experiments also confirm our associations in addition to supporting the hexamer model. We understand multiple models perhaps could predict our experimental results, especially when hexamers are also the regions that appear to primarily drive the MaxEnt scores (Supplementary Figure S16). We should however point out that we did not design the experiments looking at these parameters. For example, SpliceAI is trained based on 5Kb sequences upstream and downstream of the splice-site. Our constructs are only 100bp intron with the mCHERRY of roughly 800bp length.

4) The numbering of nucleotides around the core splice site must be altered throughout the text to conform with standard nomenclature.

We have revised and addressed this concern.

5) There needs to be a direct comparison with prior work on sQTLs in these three species, including a Venn diagram or upset plot showing which associations are shared between these studies. In particular, cite and compare Qi et al. 2022 <https://www.nature.com/articles/s41588-022-01154-4> Garrido-Martin et al. 2021 <https://www.nature.com/articles/s41467-020-20578-2>

Khokhar et al. 2019 <https://www.frontiersin.org/journals/plant-science/articles/10.3389/fpls.2019.01160/full>

We present the most or closest associated SNP only and not tens to hundreds of significant SNPs that are in the GWAS peak, all of which would be typically reported in a sQTL analysis. In addition, our phenotypes are distinctly different, and the SNP is linked to a specific site in our analysis. In contrast, in all sQTL studies, the SNP is associated with some measure of splicing at the gene level. Therefore, a direct comparison with Venn diagram or upset plot is not ideal. However, we can make some comparisons at the gene level. In revision, we have made a comparison with the GTEx sQTL data itself presented in the GTEx portal, which is based on “intron excision ratios” of Leafcutter in Supplementary Figure S3.

Are there systematic differences?

We have indeed observed massive differences.

First, most of these sQTL studies are not really genome-wide Studies (except for Khokhar et al 2021) but rather based on the assumption that most of the splicing variation is “cis” and thus they have analysed only “cis” variants [within 5 or 10 Kb (or within 2Mb in case of Qi et al) from the gene]. Therefore, there is not even a possibility of detecting “trans” associations in these studies.

Second, the smaller window used obviously reduces the number of variants tested as opposed to the purely unbiased Genome Wide analysis presented in this manuscript. Due to fewer variants being tested, the significance thresholds are adjusted in a different manner, which has resulted in calling several variants to be of significance, which in a genome-wide study simply will not be called significant. As a result, there is an increased susceptibility for spurious (false positive) associations when compared to what we have presented in this paper.

Third, in all these sQTLs there is no direct or specific link with a particular splice-site. Therefore, they simply lack the specificity that is presented in our GWAS association tables, where the variant effect is specifically linked with specific splice-sites. We have also previously shown that Khokhar et al study suffers from systemic problems, probably related to their method of measuring the phenotype (Dent, NAR-GAB, 2021).

To address this concern, we have now presented the heart sQTL analysis from GTEx consortium, based on Leafcutter intron excision ratios with SpliSER-GWAS (Supplementary Figure S3). There are 3055 genes that show significant sQTLs from Leafcutter compared to 2599 genes that we mapped via SpliSER-GWAS. There was a 40% overlap with 985 genes being common across these two approaches. For all these common genes, our SpliSER-GWAS linked variants with specific splice-sites clearly than previous sQTL analysis. For example, of the 16 splice-site mutations that were mapped specifically to the splice-site itself by SpliSER-GWAS, we only obtained a single association in the sQTL analysis. **Of the 215 SNPs that fall within the +/-50bp region of a given splice-site (i.e., likely causal), sQTL analysis picked up only 3 of these.** This analysis suggests that the phenotypes that are

used in a sQTL analysis (intron excision ratios) are less effective in picking causal mutations.

Finally, we systematically analysed all the genes that were unique to sQTL, where we failed to map any splicing variation by SpliSER-GWAS and assessed two parameters. First, whether any sites in these genes were identified to be variable (i.e., variability falls within the upper quartile) and second, if it does, how do the SpliSER plots look for those sites that are in the upper quartile, even if the sites are not the same. We observed majority of these sites to be not in the upper quartile and thus we would consider them to be not variable. Second, for those which were present, the SpliSER-GWAS plots looked with considerable background noise, thus we would not consider them to be of significance. We have now explained this along with Supplementary Figure S3.

However, looking at the distribution of p-values, one can notice that SpliSER-GWAS p-values are generally lower when compared with sQTL p-values. In particular, it appears genes that are missed by sQTL and uniquely picked by SpliSER-GWAS had a higher p-values as well as cleaner GWAS peaks. In contrast those that were not picked by SpliSER-GWAS and unique to sQTL analysis had significantly larger noise in GWAS plots and the p-values were marginal suggestive of an increased susceptibility to infer spurious associations. In summary, SpliSER-GWAS is perhaps more conservative, but generally outperformed sQTL analysis. These findings are now included in the manuscript.

6) There needs to be a discussion of the importance of LD in the interpretation of results.

We have discussed LD in the revision.

6) The authors focus on nucleotides within 100 bp. of an affected splice site (e.g. Fig. 3 and Supplementary figure 5), and present megabase scale data. However, important genetic variation has significant long-range effects within a transcript (50 nt. to 20,000 nt.). How significant is genetically associated splicing variation at this distance?

We report on a genome-wide analysis of variants associated with splicing differences and we have indeed mapped several variants at long distances, which indicate that they are important and strong enough to be mapped in a GWAS. We have presented all associations in full detail in the Supplementary Table S6.

While these are associations, causality requires further experimental analysis at an individual level. Therefore, we focused on mapped variants within 100bp since the likelihood of the identified SNP being causal is higher within 100bp compared to long distances.

To address the query, we have now presented what proportion of the variants affect non-adjacent exon/intron on the same gene suggestive of distal effects or due to alleles in LD with other closer SNPs, which we may have missed due to reduced allelic frequencies in the revision.

7) The information shown in supplemental figures S10 through S14 must be made available as supplemental data tables.

We have already presented them as supplemental Tables S8-S11.

Specific minor points:

8. the abstract should say what three species are examined.

Modified in the revision.

9. "completing" in the legend to Fig. 1

Thank you. Now changed to competing.

10. In Fig. 2, PVE runs from 0 to 1 and is therefore presumably proportion rather than percentage.

Thank you. Legend changed to reflect the same.

11. There is no scale to the Fig. 1 A and B

We have indicated that these values refer to the differences in Splice site strength (SSE), which has the range of 0 to 1 in the legend.

12. The interactions shown by outliers in Fig. 3 E and F are presented in a very confusing way. Perhaps individual points could be labelled. This will take more space, but that would be justified.

We have now changed the figure and improved the presentation and hope this is clear now.

13. Fig. 3. "highest associated SNPs" by what criterion? p value? difference in SSE?

This is based on p-value, and it is now indicated in the legend.

Reviewer #3 (Remarks to the Author):

This study combines Genome-Wide Association Studies (GWAS), and a quantitative measure of splice-site usage termed Splice-site Strength Estimate from RNA-seq (SpliSER) to investigate RNA splice-site selection across three species: *Arabidopsis thaliana*, *Drosophila melanogaster*, and humans. The main conclusion is that splice-site choice is predominantly influenced by cis-regulatory elements, especially the hexamer sequences adjacent to the splice site.

The key strength of the work is the innovative application of the SpliSER-GWAS method to examine how different genetic contexts affect splice-site selection. While GWAS has been previously applied to study splicing via splicing quantitative trait loci (sQTLs), this study is unique in treating splice-site usage itself as the phenotype. However, there are two critical issues that, in my opinion, limit the biological significance of the study.

1) The absence of replicates in the human RNA-seq data is a serious limitation. In our experience, splicing decisions supported by RNA-seq datasets can be highly variable between replicates. Therefore, the higher variation in the usage of individual splice-sites observed in the human data may be significantly affected by technical noise. Including biological replicates would enhance the reliability and robustness of the findings.

Thank you for this comment. We are unfortunately limited by the data available in the GTEx dataset and these are from multiple individuals. Therefore, to address this specific concern, we computed the correlations between specific splice-site usage data from two different unrelated tissues (Testis vs Heart) and compared these with similar tissues (skin exposed vs skin unexposed) and present this analysis in a Supplementary Figure S1. We could clearly observe clearly sites that vary between heart and testis with a correlation with a R^2 of 0.87. In contrast, the exposed and unexposed skin tissue showed a correlation with an R^2 of 0.99, which suggests that our quantifications are robust enough and unlikely to be due to technical noise. In addition, for GWAS, we use only those sites that have data from more than 100 individuals. Therefore, we believe this stringent filtering ensures the reliability and robustness of our findings through GWAS. The number of sites on the basis of which general patterns have been obtained is also substantially high. These higher numbers will offset the lack of replicates at this level of analysis. We hope this alleviates this concern about replicates.

2) The authors highlight their hexamer analysis as a novel biological observation. However, the significance of nucleotides surrounding splice sites is well established. Splice site sequences (including the hexamers in this study) have been used for over two decades to predict splice-site strength, as exemplified by tools such as MaxEntScan. The manuscript should acknowledge this existing knowledge and clarify how their approach offers new insights beyond established methods.

We agree that the significance of nucleotides surrounding splice sites is well established and our wording has provided an impression of overclaim. We have done extensive revisions to ensure that our statements are nuanced and acknowledge previous findings. We also acknowledge the predictive tools such as MaxEntScan. However, as we show in the analysis now included in the revised manuscript supplementary Figure S1, the empirical estimates differ from predicted scores, and we can obtain a ranking of the hexamers based on empirical estimates for different conditions/genotypes/perturbations etc to present a feature that is comparable across species. This computation of the empirical estimates of hexamer ranking is a novel contribution to the field and we have now clearly mentioned this throughout the text.

3) While the hexamer-based approach to predicting splice-site usage is informative, it does not account for the influence of splicing regulators. The authors begin their paper by stating that “changes in splicing can mediate phenotypic variation ranging from flowering time differences in plants to genetic diseases in humans.” This phenotypic variation often involves the regulation of alternative splicing, which is not addressed in the study.

Alternative splicing can occur due to changes in conditions, tissues or genotypes. This study addresses primarily genotype-dependent alternative splicing. This is typically relevant in the context of genetic diseases in humans or flowering time differences are caused by “genotype-induced” alternative splicing. We agree that this is one part of alternative splicing and there are tissue/organ/condition specific drivers that we are not investigating in this paper. In addition to alternative splicing at a particular locus, this study also combines splice-site usage of all sites across the genome to infer the “basic framework”, which we believe is consistent across tissue/organ/condition or even species to a certain extent. We have nonetheless addressed part of this concern by testing similarities and differences in splicing between sex (male vs female *Drosophila*), tissues (Heart vs testis in humans) and conditions (exposed vs unexposed skin tissue), included in a Supplementary Figure S1.

The authors should analyze the impact of hexamers of varying strengths in contexts where annotated splicing enhancers or inhibitors are present. Such an analysis could provide deeper insights into the role of hexamers within the complex network of splicing regulatory sequences, enhancing our understanding of how these elements interact to govern splice-site selection.

This is a great suggestion. However, even analysing hexamers in the same context already is complicated because multiple combinations to be tested require large-scale experiments we have no resources for, and we resorted to analysing the most relevant preexisting data from Rosenberg et al in Fig 4A. In addition, while the interactions are critical, they may not quite allow us to assess the intrinsic differences in hexamers, which we have analysed in this paper.

Additionally, the regulation of alternative splicing is highly tissue-specific, a factor that is ignored throughout the manuscript, particularly in the experiments involving the transfection of minigenes into HEK cells.

Tissue-specific alternative splicing is an important aspect, but not the focus of our work, which focuses on the effect of genetic/genomic variation on splicing. In our experiment, specific cases were chosen from the RNA-seq data to analyse introns that are spliced in all individuals of the analysed GTEx data (good intron-RPS10) or not spliced in any individual (bad intron-CALM3). We first tested some HEK transcriptomes to check whether HEK cells match with what we have observed in GTEx data, which confirmed for these experiments tissue-specificity in the context of testing in HEK cells is not an issue. Having established that premise, we have compared the impacts in the same context which allowed us to test the effects.

Additional comments:

4) The observation that trans variants are not prominently detected as hotspots in SpliSER-GWAS does not imply that these variants are irrelevant in splice-site selection. In fact, SpliSER-GWAS appears to be a promising tool for discerning the influence of trans variants.

We agree that SpliSER-GWAS is indeed promising to detect trans variants as opposed localised sQTL studies as commonly done in other studies. We have now included a section discussing our “trans” associations. Hot spots will likely impact

multiple splice-sites from different genes. However, we did not identify such variants in the data analysed. Without ruling out this possibility, we are simply observing that there are no hotspots whereby one variant affects multiple splice sites in different genes as expected for variants generally controlling a core component of the splicing machinery. Even this does not rule out their role but rather points to the idea that mutations at the binding sites for the RNA binding proteins evolutionarily is more tolerable and may provide a specific outcome, without compromising much of the other properties/effects as opposed to having variants in the core components themselves.

Given that the impact of cis-regional variants has already been extensively explored, both through functional studies and new machine learning algorithms, it is crucial to further investigate how trans variants might interact additively with local cis variants, including the previously mentioned hexamers, in influencing splice-site choices. This could provide a more comprehensive understanding of the genetic architecture underlying splice-site selection.

While we are mapping trans variants, we do not have enough confidence on the causality of these variants as opposed to cis variants that we map in close vicinity to the splice-site. We can certainly verify associations (which we have done in Arabidopsis), but still does not confirm causality, which requires perturbing/editing these SNPs in a systematic manner to validate the associations, which is beyond the scope of our current work.

However, in several instances, where we have mapped both “cis” and “trans” associations for the same splice-site, which is suggestive of “cis-trans/cis-cis/trans-trans” additive effects on splicing. In summary, we had 248 instances in Arabidopsis (Supplementary Table 2), and 152 (86 cis-trans +43 cis-cis +23 trans-trans) instances in Humans (Supplementary Table 4), where we could find such additive effects. For these, further experimental analysis would be required to test putative causal mechanisms.

5) The claim that the results from this method are more interpretable than those obtained with SpliceAI might not be entirely accurate. While they may indeed be more reliable, the interpretability of these results remains limited. For example, the explanation that some hexamers are stronger than others at the 5' splice site due to their resemblance to the sequence recognized by U1 snRNA—stronger hexamers more closely mimic this sequence, whereas weaker ones diverge—is a reasonable observation. However, this alone does not suffice to deem the method as highly interpretable. Further analysis and explanation are needed to fully understand the nuances of how and why certain hexamers influence splice site selection more than others.

Our interpretation is based on two aspects. First, with the machine learning approaches such as SpliceAI, the underlying logic for the predictions is not explicit. Second, the functional correlation that we observe extends beyond species. However, we have revised our statement to ensure it is accurate and not hyperbolic. While this comment is well motivated, we do find that the average hexamer strength is almost perfectly correlated with the distance to the U1 snRNA providing a biological basis for the observations. Nevertheless, we note that the hexamers

explain only a maximum of 60-70% of splice site choice, which presents sufficient opportunities such as RBPs modulating the access of the hexamer for the U1 snRNA binding, assessment of the kinetics, affecting RNA structure etc. We have now included comments on these in the revised document.

6) The supplementary tables provided at the end of the document, which detail the strength of each splice site, are indeed useful. However, creating an online tool that allows users to determine the hexamer strength of a splice site would be extremely valuable to the research community. Such a tool would not only facilitate easier access to this information but also enhance the practical application of the data in different research scenarios.

We have made the entire data available in a supplementary table and we are working to make the entire GWAS data available in an interacting manner. However, at present, we have limited resources, and interactive accessibility is not within the scope of this manuscript.

7) In this study, RNA-seq data from Arabidopsis were aligned using the TopHat2 aligner, while Drosophila and human data were processed with the STAR aligner. The choice of different aligners for different species raises questions, particularly since the central focus of the study is to understand what determines splice-site choice across different loci and species. The choice of aligner can significantly impact the mapping of spliced reads, which could influence the results. Ideally, it would be beneficial to standardize the use of aligners across all species studied to ensure consistency in data processing. Recognizing the substantial effort required to re-align the datasets, at a minimum, it would be necessary for the authors to demonstrate that both aligners perform comparably across all three organisms and across the full spectrum of splice-site strength estimates. This would help to validate the findings and confirm that the differing aligner choice does not skew the results.

TopHat2 has historically been our aligner of choice, but GTEx data was aligned with STAR, and we used the same alignments to make our analysis comparable to all previous studies and all subsequent analysis including the multi-species hexamer analysis on 25 species was all done with STAR. Both using Burrows-Wheeler approach, we have observed no effect of the aligner used (i.e., best nucleotides for splicing, general correlations of hexamers, hexamer rankings in comparison with other plants etc).

To fully address reviewers concern relating to whether the increased intronic bias that we see in Arabidopsis GWAS could be driven by differences in the aligners, we took a subset of Arabidopsis accessions and aligned them with STAR to check whether there are any systematic differences. We first assessed through a principal component analysis of the sites detected in all accessions (only these were considered in GWAS) and noticed that while the aligners clearly contributed to differences as the first principal component of the PCA (Supplementary Figure S6), all samples aligned with the same aligner clustered in the same. For each aligner splice-site strengths were distributed in the same way, hence showing same pattern irrespective of the aligner. Therefore, it appears that as long as all samples are aligned with the same aligners, it is unlikely to have an impact on the genetic mapping. To further confirm this, we looked at the correlations between same

sample/same aligner ($R^2=0.99$); diff sample/same aligner STAR ($R^2=0.97$), diff sample/same aligner TopHat2 ($R^2=0.96$), same sample/different aligner ($R^2=0.92$), diff sample/diff aligner ($R^2=0.89$) and found they all were comparable (Supplementary Figure S6).

8) "We filtered sites which had at least 10 reads crossing the splice-site in at least three replicates, in at least 100 accessions and taken them for further analysis." While this filtering criterion helps minimize false positives, it also likely reduces the detection of splice-sites that are lowly transcribed or infrequently used. This approach could significantly influence the results of downstream analyses. It would be informative to know whether the authors considered alternative thresholds. Additionally, a rationale for setting the threshold at 10 reads would help clarify the basis for this specific choice, ensuring that it optimally balances sensitivity and specificity.

We described and explained this in our previous paper (Dent et al, NAR-Genomics and Bioinformatics, 2021) and indicated that while we used 10 reads, reasonable signals in GWAS were seen even with as little as 3 reads. However, this does not particularly affect the power to detect rare splicing event since the 10 reads include not only those in which the splice-site is used, but also those in which the site could be potentially used (in other words, any reads, including where the gap is over the splice site).

For the issue of lowly transcribed genes, the number of 100 accessions considered leads to a minor allele count of at least 5 in the GWAS analysis. Reducing this number to less than 5 would lead to detecting associations that occur in less than 5 accessions, which is likely to increase spurious associations. In addition, this helped us to avoid computational overload (for example, running a Human GWAS for a single tissue on a high-performance cluster could take 20-30 days).

9) Page 7, paragraph 3: The authors should provide a detailed definition and context for what is meant by "GWAS peaks" in the study.

We have defined this now in the manuscript.

Point by point response

REVIEWER COMMENTS

Reviewer #1 (Remarks to the Author):

In revision, the authors of Dent et al., now entitled 'A basic framework to explain splice-site choice in eukaryotes', make improvements to their manuscript and provide some additional analyses which are helpful. However, some of the concerns raised by myself, and reviewers 2/3, remain.

In particular (as both myself and Reviewer 3 point out in the original review), despite the introduction & abstract highlighting regulatory examples of AS as the rationale for, and importance of, the work, the author's analysis still does not consider the tissue- context for alternative splicing (AS) signal thought to be relevant for many AS examples (as shown across decades of literature). Further, the authors still advertise the approach as a general framework to explain splicing decisions, despite this overarching blindspot (rather than a 'minimal basic framework' or other more attenuated statement).

The fact that the authors plan to focus on tissue-specific AS in another paper, does not exempt from reining in claims to only those that are justified given the data presented, or to provide additional analysis to support the broader claims of splicing generality.

Please see our responses under major comment 1 below.

While the revised version is better than the original draft, I feel that addressing the following major comments, would improve the manuscript:

1. In revision/rebuttal Dent et al. have made it clear that they are intending to quantify and explain the impact of genotype on splicing. However, by focusing their analysis primarily on heart tissue (where muscle is one of the tissues with the lower frequency of AS reported; eg. PMID:15461793), they are in effect, studying how genotype can effect otherwise constitutive splicing, or a highly limited view of the effect of genotype on AS (much of which is regulated across tissues/cell-types). The addition of one more tissue does not greatly alleviate this limitation.

The reviewer raises a valid point: since much of AS is tissue- or cell-type-specific, the use of only one tissue provides a limited view of the effect of genotype on AS. However, splice site competition itself is not restricted to AS, but all splicing (i.e., including constitutive splicing). For example, in our data presented in the manuscript ~85% of the splice-sites detected in the heart tissue have a competing site. Our interest is to look at the impact of genetic variation on “splicing” and our framework is

applicable irrespective of whether it is constitutive or alternative. When we claim that our framework explains *most* splicing choices in a given transcriptome, we are considering the usage of one site in preference to other available nearby sites, regardless of whether AS is observed. The examples we provided includes both context-dependent changes in splicing without sequence impacts (e.g., flowering time in plants) as well as sequence-dependent impacts on splicing (e.g., genetic diseases in humans, cancer). Thus, we referred to these as changes in splicing or differential splicing rather than alternative splicing throughout the manuscript.

We have revised the introduction to highlight the importance of context-specific (tissue/condition-specific) splicing regulation. We have also toned down the final paragraph of introduction specifically stating that our contribution does not explain context-specific splicing regulation.

Having clarified this, we fully understand the limitation of using a single tissue. However, there are at least four lines of evidence that support our assertions in the manuscript. First, we detect a similar number of splice-sites in heart (~450,000) which appears to be more representative of a typical tissue in terms of the number of unique splice-sites in GTEx data, and it is not an outlier (Please see the figure below shown only here for clarification and not a figure in the manuscript). While it is different from testis, which is an outlier with maximum number of splice-sites, it is like most other tissues.

Second, we showed that despite numerous splice sites having different splice-site strength estimates (SSE) between the two tissues (as seen in scatter plots of Supplementary Figure S1C & S1E), *k*-mer ranking of SSE remained generalisable between tested tissues. This is also supported by the high correlations seen in *k*-mer

rankings between tissues (including brain and heart) as shown in Supplementary Figure S20A & B.

Third, as shown by Barbosa-Morais et al as well as our own data, variation in splicing across the species, clusters by species than tissue, which suggests that tissue-specific splicing itself acts on top of a basic framework that operates at the level of species. Therefore, we believe our findings argue for a minimal/basic framework that explain most of the splicing variation across all splice-site choices (both constitutive and alternative).

Finally, our generality of the finding comes from our analysis of intronic hexamer rankings across more than 25 species, which argues for a conserved framework. Despite, different species/tissues/conditions/analysis methods, the signal from intronic hexamers still explains most variation in splicing.

The reviewer is correct in saying that our analysis does not explore splicing variation across many possible contexts (e.g., tissues). We have listed this limitation and pointed out that we cannot rule out some patterns which may become visible when multiple tissues are analysed. Future work will further uncover tissue specificity of genetic variants which control splicing. Our analysis in heart tissue has, however, discovered thousands of splicing-related variants, and our focus is to showcase general links between genetic variation and splicing which hold across many different contexts.

Moreover, the fact that the authors observe that most sQTLs reside outside of the core splicing motifs and acknowledge that these likely alter RBP-binding sites, and given that many RBPs are differentially expressed across tissues, the effect of a genotype on splicing likely differs greatly depending on the tissue at hand. For example, it is plausible that the same SNP could enhance splicing in one tissue and repress it in another. This is an important concept that is completely lacking in this paper.

Context-dependent opposite effects of the SNP are an interesting possibility. We tested this in the context of sex-specific splicing. Our preliminary findings with *Drosophila* male vs female suggest such differences are extremely rare, at least among mappable variation. Among thousands of common sites for which we could map both males and females, we did not find instances, particularly among what we would potentially consider as causal SNPs (e.g., near the splice sites), where the SNP promoted splicing in males and suppressed in females. There are certainly examples, where the SNP makes a difference in one sex, but not in another, but not examples where the SNP has an opposite effect. We have added a line in the revision when discussing context-dependent expression of RNA binding proteins.

2. In the author's rebuttal re-analysis of the data from Barbosa-Morais et al., their interpretation is flawed/incomplete, as it does not include/consider that Barbosa-Morais et al. also report higher frequency of alternative splicing in primates and higher vertebrates, and greater frequency of tissue-specific splicing in brain tissue

(nice overview here PMID:37993689). While there is also signal outside of the differentially spliced events (at given significance cutoff) that maintain species-specific clustering, that does not change the interpretation that tissue-specific splicing has more rapidly expanded/evolved in higher vertebrates (which is by definition, in a species-specific manner).

Our intention was not to refute the claims of Barbosa-Morais et al. Our aim was simply to clarify the apparent contradiction raised by the reviewer. Our observation of deep conservation of the logic of splice site choice at intronic hexamers does not contradict other reports of splicing evolving rapidly compared to gene expression. The expanded splicing also still follows the same logic of splice site choice we reported in this paper as shown by the correlations of the hexamer rankings in the same dataset (Supplementary Figure S20). In fact, both works suggest that the choices occur at the species level and we do not see any contradiction and we do not make any specific claims about rapid evolution in comparison to gene expression. We have updated the text to make this clearer.

3. I'm not sure I agree with the author's assertion that there is a difference between wanting to 'explain splicing' and 'predict splicing'. If one can provide deterministic rules or categories that 'explain' splicing outcomes, then they are in essence providing a ruleset to predict splicing. If I am understanding the rebuttal correctly, the authors suggest that their framework benefits from being simple and interpretable, rather than accurate like AI methods. But does that not suggest that scoring splice-site hexamers is overly simplistic, and does not well capture the full cis-regulatory context that is essential for splicing determination?

Yes. We agree that we are indeed providing deterministic rules that explain splice-site choices at a basic level. It is simplistic, but explains considerable fraction of the splicing variation. On the other hand, AI methods predict more of the variation, but do not provide any interpretable rules. It is on this basis we use the term 'explain' to signal to the reader that our work is distinct from a splice-site prediction tool. Please note that even in the best circumstances, hexamers do not explain all (Fig 4A), but a substantial portion of splicing variation.

Reviewer #2 (Remarks to the Author):

This revised version is acceptable with minor revisions (not requiring re-review). It is a significant and important contribution to the literature despite a tone that is dismissive of vast amounts of prior research.

Thank you for the feedback. As mentioned above we have made efforts to improve tone of the introduction and results, to avoid coming across as dismissive, which was not our intention.

Minor revisions:

1) The third sentence of the abstract ("However, how genetic variation influences splice site strength is largely unknown ...") strikes me as simply false, due to the vast literature on the impact of genetic variation on splicing. I think that what the authors mean is that splice site usage per se has not been previously used as it is here. The sentence has to be revised to make it clear what is novel and what is not.

We agree. We have revised the sentence to make this distinction clearer.

2) Similarly, "The rules that govern which GU/AG become splice site is still unclear." Should be changed to something like "... remain incompletely described."

We have corrected this sentence accordingly.

3) pg. 15 - "optimal k-mer" should be optimal "intronic k-mer"
Updated.

4) pg. 11 - "... trans associations (Table 2)" should be Table 1
Updated.

5) pg. 13 - "these patterns are driven by not evolved neutrally" is agrammatical.
We have corrected this grammar issue.

Reviewer #3 (Remarks to the Author):

The authors have substantially improved the clarity and depth of their analysis, included additional data that strengthens their conclusions, and more clearly positioned their findings within the context of existing literature. Importantly, they have also moderated some of the more speculative statements that were present in the original version.

Thank you.

While some of my original concerns have not been fully addressed, particularly in relation to potential artefacts due to lack of RNA-seq replicates and influence of splicing enhancers or inhibitors, I recognize that the revised manuscript represents a significant improvement overall. In my opinion, it will be of interest to the splicing community, as it provides a new approach to examine how different genetic contexts affect splice-site selection.

Thank you. We appreciate the positive feedback.